



# Comparison of airborne measurements of NO, NO₂, HONO, NOy and CO during FIREX-AQ

Ilann Bourgeois[1,2,*], Jeff Peischl[1,2,*], J. Andrew Neuman[1,2], Steven S. Brown[2,3], Hannah M. Allen[4], Pedro Campuzano-Jost[1,3], Matthew M. Coggon[1,2], Joshua P. DiGangi[5], Glenn S. Diskin[5], Jessica B. Gilman[2], Georgios I. Gkatzelis[1,2,a], Hongyu Guo[1,3], Hannah A. Halliday[5,b], Thomas F. Hanisco[6], Christopher D. Holmes[7], L. Gregory Huey[8], Jose L. Jimenez[1,3], Aaron D. Lamplugh[1,2], Young Ro Lee[8], Jakob Lindaas[9], Richard H. Moore[5], John B. Nowak[5], Demetrios Pagonis[1,3,c], Pamela S. Rickly[1,2], Michael A. Robinson[1,2,3], Andrew W. Rollins[2], Vanessa Selimovic[10], Jason M. St. Clair[7,11], David Tanner[8], Krystal T. Vasquez[4], Patrick R. Veres[2], Carsten Warneke[2], Paul O. Wennberg[12,13,], Rebecca A. Washenfelder[2], Elizabeth B. Wiggins[5], Caroline C. Womack[1,2], Lu Xu[12,d], Kyle J. Zarzana[1,2,e] and Thomas B. Ryerson[2,f]

[1]Cooperative Institute for Research in Environmental Sciences, University of Colorado Boulder, Boulder, CO, USA
[2]NOAA Chemical Sciences Laboratory (CSL), Boulder, CO, USA
[3]Department of Chemistry, University of Colorado Boulder, Boulder, CO, USA
[4]Division of Chemistry and Chemical Engineering, California Institute of Technology, Pasadena, CA, USA
[5]NASA Langley Research Center, Hampton, VA, USA
[6]Atmospheric Chemistry and Dynamics Laboratory, NASA Goddard Space Flight Center, Greenbelt, MD, USA
[7]Department of Earth, Ocean and Atmospheric Science, Florida State University, Tallahassee, FL, USA
[8]School of Earth and Atmospheric Sciences, Georgia Institute of Technology, Atlanta, GA, USA
[9]Department of Atmospheric Science, Colorado State University, Fort Collins, CO, USA
[10]Department of Chemistry and Biochemistry, University of Montana, Missoula, MT, USA
[11]Joint Center for Earth Systems Technology, University of Maryland Baltimore County, Baltimore, MD, USA
[12]Division of Geological and Planetary Sciences, California Institute of Technology, Pasadena, CA, USA
[13]Division of Engineering and Applied Science, California Institute of Technology, Pasadena, CA, USA

[a]now at Institute of Energy and Climate Research, IEK-8: Troposphere, Forschungszentrum Jülich GmbH, Jülich, Germany
[b]now at Office of Research and Development, US EPA, Research Triangle Park, NC, USA
[c]now at Department of Chemistry and Biochemistry, Weber State University, Ogden, UT, USA
[d]now at 1 and 2
[e]now at 3
[f]now at Scientific Aviation, Boulder, CO, USA
*corresponding authors: Ilann Bourgeois (ilann.bourgeois@colorado.edu)/Jeff Peischl (jeff.peischl@noaa.gov )





**Abstract**

We present a comparison of fast-response instruments installed onboard the NASA DC-8
aircraft that measured nitrogen oxides (NO and $NO_2$), nitrous acid (HONO), total reactive
odd nitrogen (measured both as the total ($NO_y$) and from the sum of individually measured
species ($\Sigma NO_y$)) and carbon monoxide (CO) in the troposphere during the 2019 Fire
Influence on Regional to Global Environments and Air Quality (FIREX-AQ) campaign. By
targeting smoke from summertime wildfires, prescribed fires and agricultural burns across
the continental United States, FIREX-AQ provided a unique opportunity to investigate
measurement accuracy in concentrated plumes where hundreds of species coexist. Here, we
compare NO measurements by chemiluminescence (CL) and laser induced fluorescence
(LIF); $NO_2$ measurements by CL, LIF and cavity enhanced spectroscopy (CES); HONO
measurements by CES and iodide-adduct chemical ionization mass spectrometry (CIMS);
and CO measurements by tunable diode laser absorption spectrometry (TDLAS) and
integrated cavity output spectroscopy (ICOS). Additionally, total $NO_y$ measurements using
the CL instrument were compared with $\Sigma NO_y$ (= NO + $NO_2$ + HONO + nitric acid ($HNO_3$) +
acyl peroxy nitrates (APNs) + submicron particulate nitrate ($pNO_3$)). The aircraft instrument
intercomparisons demonstrate the following: 1) NO measurements by CL and LIF agreed
well within instrument uncertainties, but with potentially reduced time response for the CL
instrument; 2) $NO_2$ measurements by LIF and CES agreed well within instrument
uncertainties, but CL $NO_2$ was on average 10% higher; 3) CES and CIMS HONO
measurements were highly correlated in each fire plume transect, but the correlation slope of
CES vs. CIMS for all 1 Hz data during FIREX-AQ was 1.8, which we attribute to a reduction
in the CIMS sensitivity to HONO in high temperature environments; 4) $NO_y$ budget closure
was demonstrated for all flights within the combined instrument uncertainties of 25%.
However, we used a fluid dynamic flow model to estimate that average $pNO_3$ sampling
fraction through the $NO_y$ inlet in smoke was variable from one flight to another and ranged
between 0.36 and 0.99, meaning that approximately 0–24% on average of the total measured
$NO_y$ in smoke may have been unaccounted for and may be due to unmeasured species such
as organic nitrates; 5) CO measurements by ICOS and TDLAS agreed well within combined
instrument uncertainties, but with a systematic offset that averaged 2.87 ppbv; and 6)
integrating smoke plumes followed by fitting the integrated values of each plume improved
the correlation between independent measurements.



## 1. Introduction

Biomass burning (BB) can take multiple forms (e.g., wildfires, prescribed fires, agricultural burns, grass fires, peat fires) and accounts for a large fraction of global carbon emissions with consequences for climate (Bowman et al., 2009; van der Werf et al., 2010, 2017) and biogeochemical cycles (Crutzen & Andreae, 2016). BB also contributes substantially to the atmospheric burden of trace gases and aerosols (Andreae, 2019), causing poor air quality on regional to continental scales (Jaffe et al., 2020; O'Dell et al., 2019; Wotawa, 2000) and posing a major threat to public health (Johnston et al., 2012, 2021). In the United States (US), wildfires mainly occur in the western states and in Alaska and burned over 4.5 million acres in 2019 (US National Interagency Fire Center, https://www.nifc.gov/fire-information). Wildfires frequency and severity are predicted to increase in response to a warmer, drier climate (Burke et al., 2021; Westerling, 2016) and also to increasing human-caused ignition (Balch et al., 2017). In comparison, prescribed fires, which are common practice in the southeastern US, burned an estimated 10 million acres in 2019, to which agricultural burns added another 2–3 million acres (Melvin, 2020). While agricultural burns are usually smaller and less intense than wildfires or prescribed fires, they occur more frequently and throughout the whole year, and can significantly impact local air quality (Dennis et al., 2002; McCarty, 2011).

Rising interest in the impact of fires on climate and air quality over the past decades has resulted in a series of laboratory studies of BB emissions in the US such as the FLAME-4 experiment in 2012 (e.g., Stockwell et al., 2014) and the FIRELAB study in 2016 (e.g., Selimovic et al., 2018). Recent, large-scale field studies such as AMMA (e.g., Liousse et al., 2010), BBOP (e.g., Collier et al., 2016) and WE-CAN (e.g., Calahorrano et al., 2020) have been dedicated to sampling and characterizing emissions and atmospheric chemistry from fires. The focus of the joint National Oceanic and Atmospheric Administration (NOAA) / National Aeronautics and Space Administration (NASA) Fire Influence on Regional to Global Environments and Air Quality (FIREX-AQ) airborne campaign was to provide comprehensive observations to investigate the impact of summer time wildfires, prescribed fires and agricultural burns on air quality and climate across the conterminous US. Accurate measurements facilitate understanding of fire emissions, processing and impacts. In situ, fast-response measurements of trace gases in the atmosphere conducted from airborne platforms provide unique data sets that enhance our understanding of atmospheric composition and chemistry. One method for evaluating measurement accuracy is by comparison of independent measurements using different techniques. A relatively small body of literature reported comparisons of methods for in flight detection of tropospheric carbon monoxide (CO) and reactive odd nitrogen species measured both as the total ($NO_y$) and from the sum of individually measured species ($\Sigma NO_y$), and these studies have shown that such comparisons are valuable for identifying instrument artifacts and quantifying measurement uncertainties (Eisele et al., 2003; Gregory et al., 1990; Hoell et al., 1987; Hoell et al., 1987; Sparks et al., 2019). During FIREX-AQ, a large suite of airborne instruments, detailed in the following sections, performed independent in situ tropospheric measurements of one or more fire-science relevant reactive nitrogen species and CO aboard the NASA DC-8 aircraft.



Additionally, FIREX-AQ provides a unique opportunity to investigate measurement accuracy
in concentrated smoke plumes where hundreds of species coexist.
Nitric oxide (NO) and nitrogen dioxide ($NO_2$) are among the largest components of the
reactive nitrogen budget emitted by biomass burning and are produced by the oxidation of
reduced nitrogen species present in the fuel in the flaming stage of combustion (Roberts et
al., 2020). $NO_x$, defined as the sum of NO and $NO_2$, directly affects atmospheric oxidation
rates and ozone ($O_3$) production within fire plumes (Robinson et al., 2021; L. Xu et al.,
2021). It also contributes to the formation of secondary aerosols and N transport and
deposition to ecosystems downwind (Galloway et al., 2003; Kroll & Seinfeld, 2008; Ziemann
& Atkinson, 2012). Therefore, two independent NO and three independent $NO_2$
measurements were part of FIREX-AQ to provide continuous in situ observations, as
described in section 2 below. Nitrous acid (HONO) is emitted directly to the atmosphere
through various combustion processes including BB. The rapid production of OH from
HONO at the early stage of smoke plume formation (Peng et al., 2020) results in rapid
initiation of photochemistry, with a strong influence on downwind chemical evolution of
smoke plumes (Robinson et al., 2021; Theys et al., 2020). Total $NO_y$ can be measured
through conversion of individual species to NO (Fahey et al., 1985). It is a more conserved
tracer for $NO_x$ emissions than $NO_x$ itself since it accounts for $NO_x$ oxidation products, and it
provides a mean to assess from a mass-balance approach the accuracy of $\Sigma NO_y$ budget
closure (Bollinger et al., 1983; Fahey et al., 1986; Williams et al., 1997). $\Sigma NO_y$ usually
consists of $NO_x$, HONO, nitric acid ($HNO_3$), nitrogen pentoxide ($N_2O_5$), peroxynitric acid
($HNO_4$), acyl peroxy nitrates (APNs), halogen-nitrogen species such as $ClNO_2$, other organic
nitrates such as alkyl nitrates, and particulate nitrate ($pNO_3$). Carbon monoxide (CO) is
emitted from incomplete combustion in fires and other sources, and is especially important
for characterizing the combustion stage of fires (i.e., flaming vs. smoldering) through the use
of the modified combustion efficiency (Yokelson et al., 1996). Due to its relatively long
chemical lifetime, CO is commonly used as a conserved tracer to account for dilution with
ambient air as smoke plumes are transported downwind, and accurate CO measurements are
necessary to better constrain emission factors (EFs) used in emission inventories.
This study builds on past airborne instrument comparisons and extends these analyses to a
new species (HONO), new measurement techniques (first airborne deployment of the NOAA
NO-LIF (laser induced fluorescence) and the NOAA CO-ICOS (integrated cavity output
spectroscopy) instruments) and new environments (concentrated fire smoke). In this paper we
present a comparison of NO, $NO_2$, HONO, $NO_y$ and CO measurements, which are
compounds of major interest for fire-related science, air quality and climate. In the first part
of this paper, we describe the FIREX-AQ campaign, the deployed instruments and the
methodology used to perform the comparisons. In the second part, we provide a detailed
instrument comparison for each species.
**2.  FIREX-AQ overview and instruments**
2.1 FIREX-AQ airborne mission



The FIREX-AQ campaign (https://www-air.larc.nasa.gov/missions/firex-aq/;
https://www.esrl.noaa.gov/csl/projects/firex-aq/) took place from July to September 2019.
FIREX-AQ included the deployment of multiple aircraft and mobile platforms over the
course of the campaign, however this study focuses on the heavily instrumented NASA DC-8
aircraft. The NASA DC-8 portion of the project achieved two flights over the Los Angeles
(LA) Basin and the Central Valley in California, 13 flights originating from Boise, Idaho, and
7 flights based out of Salina, Kansas. The flights from Boise were conducted over the
Western US to sample smoke from wildfires, while the flights from Salina focused on
agricultural and prescribed burns (hereafter referred to as eastern fires) in the Southeastern
US.

Most wildfire flights were designed to sample background mixing ratios, fresh emissions, and
aged smoke, whereas the eastern fire flights typically transected numerous fresh smoke
plumes several times each. For wildfires, the NASA DC-8 first flew upwind of the fire to
characterize ambient conditions unaffected by targeted fire emissions. Subsequent cross-wind
plume transects were conducted as close as possible to the fire to sample the emissions with
the minimal atmospheric ageing. Plume transects were designed to be perpendicular to the
wind direction and through the center of the vertical extent of the plume, terrain permitting.
The vertical structure of the plume was systematically assessed using a differential absorption
lidar during a lengthwise overpass above the plume from end to start. The aircraft transected
the smoke plume successively further downwind, at approximately 15–40 km intervals, to
characterize smoke evolution in a "lawnmower" pattern (Figure 1a). For several wildfires, the
DC-8 also executed flight transects along the plume axis, both toward and away from the fire
source. Most eastern fires sampled during FIREX-AQ did not produce plumes large enough
to enable regularly spaced plume transects. Most smoke plumes were therefore sampled
repetitively at the same location, sometimes with varying altitude and/or approach angle
(Figure 1b).

## 2.2 Instruments
### 2.2.1   Chemiluminescence (NO/NO$_2$/NO$_y$)
The NOAA CL instrument has been frequently used for both ground-based and airborne
measurements of NO, NO$_2$ and NO$_y$ and uses the CL detection of NO with O$_3$ added as
reagent gas (Fontijn et al., 1970; Ridley & Grahek, 1990; Ridley & Howlett, 1974; Ryerson
et al., 1999, 2000). NO, NO$_2$ and NO$_y$ are measured on three independent channels of the
instrument. The NO channel measures NO, the NO$_2$ channel measures the sum of NO and
photolyzed NO$_2$ as NO, and the NO$_y$ channel measures the total reactive nitrogen oxides
species reduced to NO. NO$_2$ is determined from the difference between signals from the NO
and NO$_2$ channels. Ambient air is continuously sampled from a pressure-building ducted
aircraft inlet to the instrument at a typical flow of 1045.1 ± 0.2, 1030.2 ± 0.2 and 1029.5 ±
0.2 standard cubic centimeters per minute (sccm) in flight for NO, NO$_2$, and NO$_y$,
respectively. In the NO$_2$ channel, NO$_2$ is photolyzed to NO using ultraviolet (UV) LEDs at
385 nm in a 45 cm long quartz cell (inner diameter of 1.2 cm) pressure-controlled at 209.8 ±
0.3 Torr (Pollack et al., 2011). In the NO channel, a similar quartz cell wrapped in aluminum
foil to avoid NO$_2$ photolysis and pressure controlled at 209.7 ± 0.3 Torr, ensures similar



residence time of sampled air in both channels. In the $NO_y$ channel, reactive odd nitrogen
species are first sampled through an inlet heated at $90.0 \pm 0.1°C$ then catalytically reduced to
NO on a gold tube surface heated at $300.0 \pm 0.2°C$ in the presence of added pure CO flowing
at $3.19 \pm 0.01$ sccm. Approximately 5% $O_3$ in oxygen is produced by corona discharge,
delivered at $73.80 \pm 0.02$ (NO channel), $74.11 \pm 0.03$ ($NO_2$ channel), and $74.60 \pm 0.04$ ($NO_y$
channel) sccm, and mixed with sampled air in a pressure ($8.65 \pm 0.02$, $8.79 \pm 0.02$, $8.56 \pm$
$0.02$ Torr for NO, $NO_2$, and $NO_y$ respectively) and temperature ($25.0 \pm 0.2$ and $25.1 \pm 0.2$
and $25.1 \pm 0.2°C$ for NO and $NO_2$, respectively) controlled reaction vessel. $O_3$-induced CL is
detected with a red-sensitive photomultiplier tube controlled at $–78°C$ with dry ice, and the
amplified digitized signal is recorded using an 80 MHz counter. Pulse coincidence at high
count rates was calculated after the mission by fitting an inverse function to the curve
between observed and theoretical count rates for known NO mixing ratios ranging from ppbv
to ppmv levels. Instrument calibrations were routinely performed both on the ground and
during flight by standard addition of NO from a gravimetrically determined NO in $N_2$
mixture ($1.38 \pm 0.03$ ppmv) delivered at $4.04 \pm 0.02$ (NO channel), $4.84 \pm 0.02$ ($NO_2$
channel), and $4.96 \pm 0.02$ ($NO_y$ channel) sccm. All measurements were taken at a temporal
resolution of 0.1 second (s), averaged to 1 s, and corrected for the dependence of instrument
sensitivity on ambient water vapor content (Ridley et al., 1992). Finally, $NO_2$ data were
further corrected for a 5% HONO interference due to HONO photolysis at 385 nm quantified
from theoretical calculation and confirmed in the laboratory using a HONO source described
in Lao et al. (2020). Under these conditions the total estimated 1Hz uncertainty at sea level
was $\pm (4\% + 6$ pptv), $\pm (7\% + 20$ pptv), and $\pm (12\% + 15$ pptv) for NO, $NO_2$, and $NO_y$,
respectively.

2.2.2    Laser Induced Florescence (NO)
The NOAA NO-LIF measurements were performed using a custom-built laser-induced
fluorescence instrument as detailed in Rollins et al. (2020). Air was continuously sampled
from outside the aircraft through an optical cell in the DC-8 cabin held to near 90 hPa. The
instrument utilizes a fiber laser system with a narrow-band laser tuned to a rotationally
resolved NO spectral feature near 215 nm. Rapid dithering on and off of this resonance
achieves 0.1 s measurements with a continuously monitored background to reduce
uncertainty in the instrument zero. The laser induced excitation of NO is followed by red-
shifted fluorescence which is detected by a photomultiplier tube operated in single-photon
counting mode. The laser is directed through both a sampling and reference cell in a single
pass for continuous monitoring of any changes in the instrument sensitivity due to changes in
the laser spectrum, or pressure of the optical cells. 500 ppbv of NO in air was flown at 50
sccm through the reference cell to ensure that measurements are occurring with the laser
tuned to the peak online wavelength. A constant flow of approximately 2500 sccm is
maintained within the sampling cell through the use of a custom inlet valve (Gao et al., 1999)
and the exhaust of both cells are tied together allowing for any changes in sensitivity due to
pressure fluctuations to be accounted for during data reduction. Hourly calibrations were
performed during each flight in which 2–10 sccm of 5 ppmv NO in $N_2$ mixture was added to
the sample flow resulting in mixing ratios of 4–20 ppbv. The sensitivity of the instrument
was determined using the in-flight calibrations to be typically 10 counts per second (CPS)
pptv$^{-1}$ with 10 CPS background achieving a detection limit of 1 pptv for 1 s integration. The
uncertainty of the instrument sensitivity is ± 6–9%. The effect of water vapor, which reduces
the sensitivity by quenching of the electronically excited NO, was accounted for during data
reduction using water vapor measurements provided by an ICOS instrument on the DC-8.
2.2.3    Laser Induced Fluorescence (NO$_2$)
The NASA Compact Airborne NO$_2$ Experiment (CANOE) measured NO$_2$ using non-resonant
LIF. The instrument is a modified version of a formaldehyde (HCHO) instrument (St. Clair et
al., 2019) with the excitation wavelength changed to 532 nm. The technique utilizes the
pulsed (80 kHz) output of a fixed wavelength, 2W, 532 nm laser to excite NO$_2$ molecules and
detects the resulting fluorescence with two identical detection axes consisting of a
photomultiplier tube (PMT) and optical filters that transmit > 695 nm. Delayed time gate
PMT counts are recorded at 10 Hz and a laboratory calibration, along with an intercept
determined by preflight zeroing, are used to provide 1Hz NO$_2$ data. The NO$_2$ measurement
uncertainty is estimated to be ± (10% + 100 pptv).
During FIREX-AQ, ambient air was sampled using a shared inlet that provided a large (10–
25 standard liter per minute (slpm)) bypass flow to the instrument rack. The CANOE
instrument pulled its 750 sccm sample flow from a shared manifold at the instrument rack.
An inline particle filter on the sample line prevented laser scatter by fine aerosol that were
not removed by the particle-rejecting inlet. A manual three-way valve outside the instrument
was used to sample from a scrubber (Drierite/molecular sieve) and provide a zero before and
periodically during the flight. Pressure in the CANOE detection cell was maintained at 40
Torr by a pressure controller that precedes the cell in the flow path.
2.2.4    Cavity Enhanced Spectroscopy (NO$_2$/HONO)
NO$_2$ and HONO were also measured by the NOAA airborne cavity enhanced spectroscopy
(ACES) instrument. This technique is based on incoherent broadband cavity enhanced
spectroscopy (CES, Fiedler et al., 2003). The CES instrument is described in full detail by
Min et al. (2016) with only minor changes for FIREX-AQ. Briefly, the system consists of
two parallel 45 cm optical cavities capped by highly reflective mirrors, with reflectivity
curves centered at 365 nm (R = 0.99987) and 455 nm (R = 0.99992). Each cavity is
illuminated by a broadband LED light source (centered at 365 and 455 nm respectively)
collimated by an off-axis parabola, and passively coupled into the cavity. The light makes
many passes before exiting the cavity into a fiber optic cable, which transmits the light to a
grating spectrometer spanning 350–475 nm. The LEDs are modulated on for 0.4 s and off for
0.08 s for charged-couple device (CCD) readout, giving a total integration time of 0.48 s per
light intensity spectrum. An absorption spectrum of the ambient air sample is determined
using the procedure presented by Washenfelder et al. (2008). The procedure requires
comparing the measured light intensity spectrum to a background spectrum of the cavity
filled with zero air, which is determined here every 10 minutes. The mirror reflectivity is
measured every hour using the Rayleigh scattering difference between helium and zero air,
and the spectrometer dark counts and wavelength calibration are measured every two hours.
A small flow from a mixture of 25 ppm $NO_2$ in air is diluted into the cavity every hour,
resulting in $NO_2$ concentrations between 50 and 100 ppbv, to assess the $NO_2$ spectral retrieval
features on the spectrometer. The absolute concentration was not used for calibration of the
$NO_2$ response, but rather for providing a reference $NO_2$ spectrum. Glyoxal reference spectra
was obtained by bubbling zero air through a Teflon bubbler with 40% glyoxal in water as in
Min et al. (2016).
Ambient air is pulled through the inlet into the two optical cavities at a flow rate of 5.4
volumetric liters per minute per cavity by a scroll pump. The air passes through two 1 μm
pore size Teflon filters before entering the instrument to remove any aerosol particles. Mirror
cleanliness is maintained by flowing 150 sccm zero air over each mirror to prevent
condensation of semi-volatile species. A pressure controller consisting of a Teflon orifice and
a variable flow to a bypass maintains the internal pressure at one of two pressure set points:
400 mbar when the aircraft was below 7.3 km, and 150 mbar above 7.3 km. The residence
time of the air inside the optical cavities is estimated to be 0.5 s.
The measured absorption spectrum is fit to a linear combination of literature or reference
spectra of absorbing gas-phase species and a polynomial to account for drifts in the cavity
stability or light source intensity, as detailed by Min et al. (2016), using a Levenberg-
Marquardt least-squares fitting algorithm. For the 365 nm channel, those species are $NO_2$,
HONO, $O_4$, and a 4th order polynomial. For the 455 nm channel, those species are $NO_2$,
glyoxal, methylglyoxal, $H_2O$, and $O_4$, as well as a 0th order polynomial, though only $NO_2$ is
presented here. The algorithm uses reference spectra for $NO_2$ and glyoxal, as measured in the
field, scaled linearly to the literature spectra of Vandaele et al. (1998) at 296 K and Volkamer
et al. (2005) at 294 K, respectively. The literature spectra from Stutz et al. (2000), Meller et
al. (1991), Harder & Brault, (1997), and Keller-Rudek et al. (2013) are used for HONO,
methylglyoxal, $H_2O$, and $O_4$, respectively. The fitting range was 438 – 467 nm for the 455
nm channel, and 362 – 387 nm for the 365 nm channel. No structure was observed in the fit
residuals. Because the 455 nm channel has higher precision, only those $NO_2$ data are
presented here, although the two channels agree to within 3%. The data are averaged to 1 s.
The reported uncertainties are ± (9% + 0.6 ppbv) for HONO and ± (5% + 0.26 ppbv) for
$NO_2$.
2.2.5    Iodide-Adduct Chemical Ionization Mass Spectrometry (HONO)
HONO was measured using a modified commercial time of flight chemical ionization mass
spectrometer (TOF CIMS, Aerodyne Research, Inc.; Lee et al., 2014; Veres et al., 2020).
Trace gases are ionized by mixing ambient air with reagent ions made in flight, and the
resulting product ions are detected. Ions are separated by mass-to-charge ratio (m/z) using a
time-of-flight mass spectrometer with a resolving power of 5000 m/Δm and a range of mass
to charge ratio up to 494 m/z. Spectra were obtained at a 25 kHz repetition rate, and then
averaged to 1 s. High resolution peak fitting was performed on the spectra, using over 500
known masses. Reagent ions were formed by flowing 1 slpm $N_2$ through a temperature
controlled $CH_3I$ permeation tube followed by a 20 mCi [210]Po radioactive source. Two reagent





ions are generated: Iodide ions (I⁻) are formed in the radioactive source, and iodide-water
clusters (I⁻•H$_2$O) are formed when I⁻ reacts with water in the ion-molecule reactor (IMR). In
the IMR, the reagent ions cluster with analyte gases to form a stable iodide adduct. The IMR
was controlled at 40 mbar pressure to reduce the effects of secondary ion chemistry that
increase at higher pressures.
Ambient air was sampled through a mass flow controlled (6 slpm) heated perfluoroalkoxy
(PFA) inlet (70 cm length, 0.64 cm inner diameter). A pressure control region upstream of a
critical orifice at the entrance to the IMR was maintained at 140 mbar, so that a constant flow
of 1.2 slpm ambient air entered the IMR to mix with the 1 slpm ion source flow. A small
nitrogen flow of about 20 sccm containing water vapor was added directly into the IMR
region and controlled to maintain a measured I⁻•H$_2$O:I⁻ cluster ratio of 50 ± 2%, in order to
maintain constant detection sensitivity. The reagent ion signals during FIREX-AQ were
typically 2 MHz for I⁻•H$_2$O and 4 MHz for I⁻, and they were stable as a function of aircraft
altitude. In the most concentrated fire plumes with CO over 7 ppm, the abundance of
reactants reduced the reagent ion signals by up to 15%. The product cluster ions were
normalized by the iodide signals to account for changes in reagent ions. The instrument
background signal was determined inflight by overflowing the inlet with scrubbed ambient
air for 30 seconds every 10 minutes through a port located 2 cm downstream of the inlet
entrance. Calibrations with Cl$_2$ and HNO$_3$ permeation sources were performed hourly in
flight to diagnose the stability of instrument sensitivity.
HONO was detected as a cluster with I⁻ that has a mass to charge ratio of 173.90575 m/z.
Contributions from the $^{13}$C isotope of formic acid at 173.91342 m/z are not completely mass
resolved but are accounted for using high resolution peak fitting and isotope ratios based on
the formic acid signal at its most abundant isotope. We know of no other contributions to the
signal at the mass used for HONO detection, consistent with previous studies (Neuman et al.,
2016). The background HONO signals were typically equivalent to a mixing ratio of 40 ppt,
and these were subtracted from the total signal to determine ambient HONO. Sensitivity to
HONO was determined in the laboratory, using a tunable, calibrated HONO source that uses
HCl reactions on humid NaNO$_2$ to generate HONO (Lao et al., 2020). The output was
calibrated spectroscopically using the NOAA ACES instrument (Min et al., 2016). The
absolute sensitivity to HONO was 3.4 ion counts/s/pptv for typical conditions. Sensitivities
normalized by the reagent ions are used to determine mixing ratios from the normalized
product ion signals. The HONO measurement uncertainty at fixed temperature was ± (15% +
3 pptv), where the first term was from the laboratory calibrations and the second was the
variability of the in-flight background determinations. The HONO measurement precision
was ±2 pptv for 1 second data. Calibrations and field work conducted subsequent to FIREX-
AQ identified a temperature dependence to the CIMS calibration.  Section 3.3 below
describes this sensitivity in more detail.

2.2.6    ΣNO$_y$



To determine the extent of budget closure for reactive odd nitrogen species during FIREX-
AQ, we compare measured $NO_y$ (see section 2.2.1) with $\Sigma NO_y$ defined as:
$\Sigma NO_y = NO_x + HONO + HNO_3 + pNO_3 + APNs$     (Eq. 2)
Other nitrogen oxides were also measured during FIREX-AQ but were not included in this
equation as they contributed on average less than 7% to the $NO_y$ budget (see section 3.4).
Further, including these measurements would have decreased data availability for comparison
with the total $NO_y$ measurement by more than 60%. These minor $NO_y$ species are alkene
hydroxy nitrates, nitromethane ($CH_3NO_2$), $N_2O_5$, and $C_1$–$C_5$ alkyl nitrates. $ClNO_2$ was also
measured by $I^-$-CIMS but not included in this work as its contribution to the $NO_y$ budget was
negligible during FIREX-AQ.
•   $HNO_3$ observations were made by the California Institute of Technology Chemical
Ionization Mass Spectrometer (CIT-CIMS) compact time-of-flight (cToF,
TofWerk/Caltech) sensor using $CF_3O^-$ ion chemistry (Crounse et al., 2006). In short,
a large flow of ambient air was rapidly brought into the aircraft through a Teflon
coated glass inlet (warmed slightly above ambient temperature), where it was
subsampled, diluted with dry $N_2$, reacted with $CF_3O^-$, and underwent subsequent
product ion analysis by time-of-flight mass spectrometry. The $HF \cdot NO_3^-$ (m/z 82)
product ion is used to quantify $HNO_3$. The hydroxy nitrates produced from the
oxidation of isoprene, ethene, propene, and butene are detected as cluster ions. In-
flight instrumental zeros were performed every ~15 minutes using dry $N_2$ and ambient
air passed through $NaHCO_3$-coated nylon wool. Laboratory-generated, water-
dependent calibration curves were performed to produce ambient mixing ratios from
raw signals. Continuous $HNO_3$ data, with the exception of zero and calibration
periods, are reported with 1Hz frequency with an uncertainty of $\pm$ (30%+ 50 pptv).

•   Particulate nitrate ($pNO_3$) was measured with a high-resolution time-of-flight AMS
(HR-AMS, Aerodyne Research, Inc., Billerica, MA, USA). The HR-AMS measured
submicron ($PM_{0.9}$; calibrated in the field as described in Guo et al., 2021) aerosol
composition at high time resolution (0.1–1 s) by flash vaporization of the aerosol, 70
368        eV electron ionization of the volatilized gas phase and subsequent analysis by mass
spectrometry (Canagaratna et al., 2007; DeCarlo et al., 2006). $pNO_3$ is detected in the
HR-AMS as the sum of $H_xNO_y^+$ ions (mostly $NO^+$ and $NO_2^+$). Typical 1 s detection
limits for $pNO_3$ were about 90 ng $sm^{-3}$ (30 pptv) for urban/background conditions.
Given the size cut in the HR-AMS instrument, $pNO_3$ does not include coarse nitrate
from the reaction of $HNO_3$ with sea salt or dust aerosol. It does include particulate
organic nitrates ($pRONO_2$; Day et al., 2021; Farmer et al., 2010), which are speciated
using the algorithm described in Fry et al. (2013) and Day et al. (2021). Likewise,
particulate aryl nitrates such as nitrocatechol also contribute to the total $pNO_3$ signal
(Guo et al., 2020). Nitrocatechol was also characterized by extractive electrospray
ionization time-of-flight mass spectrometry (EESI-MS; Pagonis et al., 2021) and





positive matrix factorization and tracer analysis suggests that total aryl nitrates could
be 3–7 times the concentration of nitrocatechol.

• APNs were measured using a thermal dissociation – chemical ionization mass
spectrometer (TD-CIMS) method. The CIMS instrument used during the FIREX-AQ
campaign was similar to that described in Slusher et al. (2004) and Lee et al. (2020).
Briefly, ambient air is sampled into the TD-CIMS through heated Teflon tubing at a
temperature of approximately 150°C to thermally dissociate APNs. The thermal
dissociation region was maintained at a constant pressure of 60 torr using a
commercial pressure controller (MKS 640) to minimize negative interference due to
NO, $NO_2$ and radical-radical reactions. In-flight calibrations were performed by
continuous addition of isotopically labeled peroxyacetyl nitrate (PAN) standard
quantified as acetate ion (61 m/z; $C^{13}H_3C^{13}(O)O^-$) in the TD-CIMS. NO was
periodically added to the inlet (~10 ppm) to react away peroxyacyl radicals and thus
to measure the instrument background signal.

• Nitromethane ($CH_3NO_2$), along with other volatile organic compounds (VOCs), was
measured by proton-transfer-reaction time-of-flight mass spectrometry (PTR-ToF-
MS; Gkatzelis et al., in prep). The PTR-ToF-MS sampled VOCs at 5Hz through short
(1 m) heated inlet. Periodically, instrument backgrounds were determined by passing
ambient air through a platinum catalyst heated to 350°C. The instrument response to
VOCs was calibrated by gravimetrically prepared standards or by liquid calibration,
as described by Gkatzelis et al. (2021). $CH_3NO$ mixing ratios were determined by
liquid calibration with an uncertainty of 30%.

• $N_2O_5$ was detected as a cluster with $I^-$ at mass 234.88574 m/z. Sensitivity was
determined by standard addition laboratory calibrations, with $N_2O_5$ generated by
reacting a $NO_2$ calibration standard with $O_3$ (Bertram et al., 2009), and quantified
using cavity ring down $NO_y$ measurements (Womack et al., 2017). For typical
operating conditions during FIREX-AQ, $N_2O_5$ sensitivity was 70 ion counts/s/ppt.
$N_2O_5$ was measured with ± (15% + 2 pptv) accuracy and 0.1 pptv precision for 1
second data. Iodide ions cluster with a DMS oxidation product, hydroperoxymethyl
thioformate (HPMTF), that has a mass only 0.0074 amu greater than $N_2O_5$, and these
two molecules cannot be completely resolved spectrometrically with the resolution
(m/Δm = 5000) of this instrument (Veres et al., 2020). For these measurements over
the continent, the contribution from HPMTF to the signal at the iodide $N_2O_5$ cluster is
assumed to be negligible.

• $C_1$–$C_5$ alkyl nitrates were measured by the NOAA integrated whole air sampling
system with off-line analysis by gas chromatography-mass spectrometry (iWAS/GC-
MS as described in Lerner et al. (2017)). There were 142 iWAS samples collected
over the LA Basin with an average fill time of 5.2 ± 0.7 seconds. There were 897
wildfire samples and 467 eastern fire samples with average fill times of 7.6 ± 1.1 and



4.5 ± 0.8 seconds, respectively. Due to the relatively fast fill times and targeted, on-
demand sampling capabilities of the iWAS, 88% and 74% were "full smoke" samples
for wildfire and eastern fire samples, respectively. All samples were analyzed in the
NOAA Chemical Science Laboratory within 213 hours of sample collection with an
average sample age of 87 ± 34 hours between sample collection and sample analysis
for FIREX-AQ.
### 2.2.7   Integrated Cavity Output Spectroscopy (CO)
CO was measured using a modified commercial off-axis ICOS instrument (Los Gatos
Research (LGR) $N_2O$/CO-30-EP; Baer et al., 2002) at approximately 4.6 µm. The
commercial instrument has two flow paths, a slow flow path with cavity pressure controlled
by an internal proportional valve, and a parallel high flow path with a needle valve to control
pressure. The instrument was modified to use only the high flow path, but with an automatic
cavity pressure controller. The needle valve was removed from the flow path in favor of a
Piezo proportional valve (Horiba Stec UR-Z732M) located near the inlet.
Air was sampled from a ram-air intake inlet through 0.64 cm (outside diameter) stainless
steel tubing. Cavity pressure was maintained at 85.0 ± 0.2 Torr in flight. Immediately inside
the fuselage, two CO (and $N_2O$) calibration gas standards known to within ±0.4 ppb CO were
regularly delivered to the inlet line during flight to evaluate instrument sensitivity between
58.4 and 993.3 ppb CO (all ICOS-CO mixing ratios are reported as dry air mole fractions).
The calibration standards were added to displace ambient air and overflow the inlet, and were
calibrated before and after the project using standard tanks tied to the World Meteorological
Organization CO_X2014A scale from the NOAA Global Monitoring Laboratory (Hall et al.,
2007; Novelli et al., 1991). The 1-sigma variability of the slope and intercept of all in-flight
calibrations was 0.6% and 0.9 ppb, respectively. A third calibration standard, referred to as a
"target" (Peischl et al., 2010), was regularly introduced to the inlet between calibrations and
treated as an unknown to evaluate long-term instrument performance. The retrieved value of
109 in-flight targets during FIREX-AQ was 301.6 ± 1.0 ppb CO compared with the
calibrated value of 301.1 ± 0.4 ppb. The precision of the measurement in flight is estimated to
be 0.4 ppb.
After the campaign, the $H_2O$ measurement was calibrated using a MBW 373LX chilled-
mirror hygrometer (MBW Calibration AG; Rollins et al., 2020). The $H_2O$ measurement is
estimated to have an uncertainty of ± (50 ppmv + 4%), and was used to convert the CO
measurement to a dry air mole fraction. The uncertainty of the dry air mole fraction of CO is
estimated to be ± (2.0 ppb + 2%) for mixing ratios below 1 ppm.
### 2.2.8   Tunable Diode Laser Absorption Spectroscopy (CO)
Carbon Monoxide (CO) was measured by tunable diode laser absorption spectroscopy
(TDLAS) using the DACOM (Differential Absorption Carbon monOxide Measurement)
instrument (Sachse et al., 1987). The TDLAS instrument configuration used during FIREX-
AQ also included channels for measurements of methane ($CH_4$) and carbon dioxide isotopes
($^{12}CO_2$ and $^{13}CO_2$). This instrument utilizes three single-mode tunable diode lasers, with CO





measured using a quantum cascade laser (QCL) at approximately 4.7 μm. The three
individual mid-infrared laser beams were combined by the use of dichroic filters and directed
through a small volume (0.3 liter) Herriott cell enclosing a 36-meter optical path. After
exiting the Herriott cell, the beams were spectrally separated and directed to individual
HgCdTe (MCT) detectors.
The lasers were operated in a wavelength-modulated mode, each at an independent
frequency, and line-locked to the centers of the species' selected absorption lines. Lines were
selected to provide both good sensitivity and good isolation from any potential spectral
interferences. Detector signals were demodulated at twice the lasers' modulation frequencies
(2F detection), and normalized by average detected laser intensity.
Ambient air was sampled through an inlet probe, compressed, and passed through a
permeable membrane dryer to remove water vapor prior to being introduced into the Herriott
cell. Due to the need for very fast time response during FIREX-AQ, the instrument was
operated with a flow of approximately 14 slpm with the Herriott cell at a pressure of
approximately 67 mbar. The resulting time response, verified with a fast-acting valve, was
faster than 0.2 s. Data were reported at both 0.2 s and 1 s timesteps.
The TDLAS instrument was calibrated using the same gas standards as for the ICOS
instrument, nominally with a 4-minute period, but often advanced or delayed in time to avoid
calibrating during fire plume encounters. Calibrations provided both slope and intercept
values tying signals to species concentrations. The very large CO concentrations encountered
necessitated post-campaign correction calibrations to account for response nonlinearity.
Post-campaign analysis of the TDLAS CO data indicated that measurement precision (1$\sigma$)
was approximately 0.1% at 1 s and 0.14% at 0.2 s. Accuracy was dependent on CO mixing
ratio, and varied from 2% to 7%.

2.2.9   $H_2O$
$H_2O$ was measured using the NASA diode laser hygrometer, an open-path infrared absorption
spectrometer that uses a laser locked to one of three water vapor absorption features near
1.395 μm, depending on the abundance of water vapor (Diskin et al., 2002; Podolske et al.,
2003). $H_2O$ mixing ratios were determined with an uncertainty of 5%.

2.2.10  Smoke age
The age of smoke from emission to sampling by the aircraft was determined from an
ensemble of upwind trajectories from the aircraft (Holmes et al., 2020). Trajectories were
computed with HYSPLIT (Stein et al., 2015) using three meteorological datasets (HRRR,
NAM CONUS Nest, and GFS 0.25°). In each of the three trajectories, the advection time was
determined from the point where the trajectory most closely approached the source fire. The
age also includes plume rise time from the surface to trajectory altitude, which was estimated
with a mean rise time of $7 \pm 4$ m s$^{-1}$ (Lareau et al., 2018). Trajectories and ages that were
grossly inconsistent with smoke transport patterns seen in geostationary satellite images were





excluded from further analysis. The ensemble of age estimates was then averaged to provide
a best estimate of smoke age. The median uncertainty in smoke age is about 27%, as
determined by spread among the ensemble of estimates.

2.3 Methodology
This study focuses on comparing the different techniques used for the measurements of one
or several reactive nitrogen species as well as CO during FIREX-AQ. Here we compare both
archived 1 s data ([https://www-air.larc.nasa.gov/missions/firex-aq/index.html](https://www-air.larc.nasa.gov/missions/firex-aq/index.html)) and the plume-
integrated data. Plume-integrated data are obtained from integrating the 1Hz data of a given
measurement over a smoke plume transect. A smoke plume transect was identified using the
time period between a CO and/or black carbon (BC) increase above a local background value
(beginning of the plume transect) and the CO and/or BC decrease back to a background value
(end of the plume transect). Background values on either side of a plume were different for
some fires in spatially heterogeneous source regions. Note that any 10 s period of background
air, even if experienced during a single smoke plume transect, was sufficient to mark the end
of one transect and the start of the next. All 1Hz data were time-aligned prior to comparison
by synchronizing features in the time series of each species. Time shifts were typically less
than 4 seconds. Some disagreement between measurement techniques is expected due to the
rapid variations sampled during FIREX-AQ, particularly when those variations occur faster
than the measurement period and/or with greater spatial heterogeneity than the distance
between the sampling locations on a large aircraft that can reach 25m in some cases.

We first calculated the slope of the linear least-squares (LLS) orthogonal distance regression
(ODR) to characterize the percent difference between measurements of a pair of instruments
weighted by the inverse of the instrument precision. Here, we used a mixing ratio-
independent instrument precision that corresponded to the $1\sigma$ precision in clean air.
Weighting the fit by this term, rather than a more accurate but labor-intensive mixing-ratio-
dependent precision, tend to overweight the highest measured mixing ratios. The slope and
intercept resulting from the ODR regression analysis provide a measure of systematic or
species-dependent instrumental biases. Additionally, we calculated the difference between a
given pair of measurements. The difference, noted $\Delta Y_{X1-X2}$ where X1 and X2 are the two
measurement techniques for detection of the Y species, provides an understanding of the
temporal evolution and environmental dependency of instrumental discrepancies. Note that
the regression analysis yields slightly different information than the calculation of the
difference: while the former is weighted more by fire plumes, where mixing ratios were
greatest, the latter is weighted more by background conditions, where most of the
measurements took place. Unless specified otherwise, all data available (i.e., both
background and fire smoke data) were included in the following comparisons. We also
calculated the fractional error (FE = $\Delta Y_{X1-X2}/Y_{avg}$ where $Y_{avg} = (Y_{X1} + Y_{X2})/2$) between pair
of instruments using specifically fire smoke data to minimize measurements below
instrument detection limits (Figures S1 and S2).

**3   Flight data comparisons**
3.1 NO



### 3.1.1 Campaign-wide comparison

The 1Hz data comparison between the CL and LIF instruments is shown in Figure 2. The overall comparison slope (± combined instrument uncertainties) is $0.98 \pm 0.08$ ($R^2 = 0.93$) with an intercept of $-2 \pm 0$ pptv (Figure 2a). Figures 3a and 4a show the two instruments' response in smoke from a wildfire and an eastern fire, respectively. While the NO signals track each other remarkably well, there is a difference in time response that is typical of the entire campaign. Figure S3 shows an expanded view of 10Hz NO and CO measurements in a partial smoke plume transect, including the transition from smoke to background air sampling. The NO signal in the CL instrument exhibits less structure than in the LIF instrument and a tail following the plume-to-ambient air transition. These tails were commonly observed during this transition. This effect in the CL instrument may partly explain the elevated scatter below the 1:1 line in Figure 2b. Integrating the NO signal across plume passes reduces the scatter due to different instrument time response: the regression analysis of smoke plume-integrated NO mixing ratios yields a slope of 0.99 ($R^2 = 0.95$) for the whole dataset (Figure 2c).

A histogram of the absolute difference between LIF and CL ($\Delta NO_{LIF-CL}$) is shown in Figure 5a. 90% of the values were between $-44$ and $43$ pptv, and the whole dataset is normally distributed around $0 \pm 0$ pptv (central value of the Gaussian fit and standard deviation). $\Delta NO_{LIF-CL}$ exhibits no significant correlation with NO and $H_2O$ mixing ratios, which suggests that there was no systematic bias between the two instruments over a wide range of NO mixing ratios and environmental conditions (Figures S4a and 6a). Similar slopes and intercepts were obtained when separately comparing NO measurements during the wildfire, eastern fire, and LA Basin sampling periods (Figures 2b and S5).

### 3.1.2 Literature aircraft NO measurement comparisons

Overall, the comparison between the two NO instruments shows an agreement within stated uncertainties. While the single-photon LIF detection of NO is a new technique that was evaluated for the first time during FIREX-AQ (Rollins et al., 2020), there are several studies that compared CL detection of NO to other measurement techniques during airborne field campaigns. The Global Tropospheric Experiment Chemical Instrumentation Test and Evaluation (GTE-CITE) was designed in the 1990's to intercompare airborne measurement techniques for trace species including NO, $NO_2$ and CO. Comparison of two CL instruments and a two-photon LIF instrument showed agreement when NO mixing ratios were higher than 50 pptv, but pointed out periods of disagreement when NO mixing ratios were lower than 20 pptv (Gregory et al., 1990; Hoell et al., 1987). The Deep Convective Clouds & Chemistry (DC3) experiment in 2012 allowed for side-by-side comparison of instruments aboard two aircrafts at two level flight legs (7 and 12 km) for flight periods spanning 20–30 minutes. Pollack et al. (2016) showed that these NO measurements from two CL instruments agreed within 2% for NO mixing ratios up to 1 ppbv. More recently, Sparks et al. (2019) reported an intercomparison of several $NO_y$ species measurements, including NO, from the Wintertime Investigation of Transport, Emissions, and Reactivity (WINTER) airborne experiment over the Northeast US in 2015. During WINTER, NO measured by CRDS and CL differed on average by 16 % across all flights, which is outside of the combined





instrument uncertainties. CL measurements were more consistent with an independent
calculation of NO based on a photostationary state assumption.
3.2 NO$_2$
3.2.1    Campaign-wide comparison
Three instruments measured NO$_2$ mixing ratios during FIREX-AQ using CL, CES and LIF
detection techniques. The 1Hz data comparison between all three instruments is shown in
Figure 7. We find that the LIF and CES overall comparison yields a slope (± combined
instrument uncertainties) of 1.03 ± 0.08 (R$^2$ = 0.98), well within the combined instrument
uncertainties of 8% (Figure 7c). However, we find that comparing either the LIF or CES
instruments to the CL instrument results in correlation slopes (± combined instrument
uncertainties) ranging from 0.88 ± 0.12 to 0.90 ± 0.11 (R$^2$ = 0.97), comparable to the 8–11%
combined uncertainties for each pair of instruments (Figures 7a and b). The higher NO$_2$
mixing ratios measured by the CL instrument are further illustrated in the time series in
Figures 3b and 4b, and is consistent with a calibration error in one or all instruments, or an
interference from another species in the CL instrument. HONO is a known source of
interference in measured NO$_2$ by instruments that use photolysis in the near-UV region
(Pollack et al., 2011). However, this interference was determined to be low (less than 5% of
HONO concentration) following laboratory tests using a HONO calibration source (Lao et
al., 2020), and the NO$_2$ measurement by CL was corrected for it. Additionally, we did not
find a correlation between either ΔNO$_{2CES-CL}$ or ΔNO$_{2LIF-CL}$ and HONO mixing ratios. There
was better agreement between the CL and the other two instruments when sampling the
wildfires (slopes of 0.91) than the eastern fires (slopes of 0.75 and 0.87 for the LIF and CES,
respectively) (Figures 7d and e). Similarly, the agreement between the CES and the LIF
instruments was near perfect during the first period (slope of 1.00), but worse during the
latter period (slope of 1.13; Figure 7f). Note that the LIF instrument did not report data for
three flights out of seven during the eastern fires sampling period. The increased difference
may be caused by the physical distance between instrument inlets combined with higher
spatial heterogeneity of trace gases in the smaller and thinner eastern fire plumes, although
higher mixing ratios of a potential interferent may still exist. Non-acyl peroxynitrate species
such as pernitric acid (HO$_2$NO$_2$) and methyl peroxy nitrate (MPN) can be abundant in smoke
plumes and interfere with NO$_2$ measurements (Browne et al., 2011; Nault et al., 2015). This
interference is the result of the thermal dissociation of HO$_2$NO$_2$ and MPN in heated inlets and
sampling lines, and impact differently each instrument depending on their flush time. During
FIREX-AQ, the CES and CL instruments had similar flush time of about 750ms meaning that
the thermal decomposition of non-acyl peroxynitrates is unlikely to explain the 10–12%
higher NO$_2$ signal in the CL instrument. Nitrated phenolic compounds can be abundant in
aged smoke (Decker et al., 2021), and have large UV cross sections (Chen et al., 2011). They
are unlikely to contribute to the interference as their NO$_2$ photolysis quantum yields are very
low. Nevertheless, further laboratory work on the NO$_2$ interference of such species in
photolytic converters is of interest. The agreement between all three instruments for
individual flights was generally within combined instrument uncertainties, but with some
variability (Figures S6–S8).





Histograms of the absolute difference between CES, LIF and CL ($\Delta NO_{2LIF–CL}$, $\Delta NO_{2CES–CL}$
and $\Delta NO_{2CES–LIF}$) are shown in Figures 5b–d. 90% of $\Delta NO_{2LIF–CL}$, $\Delta NO_{2CES–CL}$ and $\Delta NO_{2CES–}$
$_{LIF}$ values were between –298 and 338 pptv, –469 and 302, and –576 and 393 pptv,
respectively, and all are normally distributed around the central value of the Gaussian fit of
$0.038 \pm 0.001$, $-0.052 \pm 0.001$, and $-0.071 \pm 0.001$, respectively. $\Delta NO_{2LIF–CL}$, $\Delta NO_{2CES–CL}$
and $\Delta NO_{2CES–LIF}$ exhibit no significant trend with $H_2O$ mixing ratios (Figures 6b–d), yet
$\Delta NO_{2LIF–CL}$ and $\Delta NO_{2CES–CL}$ were weakly ($R^2 = 0.36$ and 0.31, respectively) correlated with
the absolute $NO_2$ mixing ratio (Figures S4b and d).

3.2.2     Literature aircraft $NO_2$ measurement comparisons
Previous comparisons of $NO_2$ airborne measurements often show periods of disagreement
between instruments, although there were some occasions where instruments agreed within
stated uncertainties. During the GTE-CITE experiment, the comparison of $NO_2$
measurements using a two-photon NO LIF system with laser photolysis of $NO_2$ to NO with a
CL detector equipped with a xenon arc lamp for $NO_2$ photolysis into NO showed agreement
within 30–40% (Gregory et al., 1990). Pollack et al. (2016) showed that two $NO_2$
measurements, both using CL but each in a different aircraft, agreed within 28% during the
DC3 campaign. During WINTER, $NO_2$ measurements by CRDS and LIF agreed with an
average proportional bias of 2% across all flights – well within combined uncertainties
(Sparks et al., 2019). During SENEX, three techniques were used to measure $NO_2$: a CRDS
instrument, a CES instrument and a CL instrument. The agreement between CRDS and CES
measurements with the CL technique was on average 6 and 10% (Warneke et al., 2016).

3.3 HONO
3.3.1     Campaign-wide comparison
The 1Hz data comparison between the CES and the CIMS instruments is shown in Figure 8,
and timeseries of HONO measurements in wildfires and eastern fires are shown in Figures 3c
and 4c, respectively. The correlation between the CES and CIMS was very high in each
plume transect (Figures 3c and 4c), but the overall comparison yielded a slope (± combined
instrument uncertainties) of $1.80 \pm 0.16$ ($R^2 = 0.77$) and an intercept of $-0.12 \pm 1.10$ ppbv
(Figure 8a). Integrating across plume transects yielded a slope of $1.34 \pm 0.16$ (Figure 8c). The
CIMS consistently reported less HONO than the CES in smoke plumes, and the average
slope between the two measurements was considerably greater during the eastern fires
compared to the wildfires (Figures 8b and S9). However, flight averages of the absolute
difference between the two measurements ($\Delta HONO_{CES–CIMS}$) ranged between –332 and 245
pptv throughout the campaign and were similarly scattered around zero during the two
different time periods (Figure S9). A histogram of $\Delta HONO_{CES–CIMS}$ is shown in Figure 5e.
90% of the values were between –965 and 880 pptv, and the whole dataset is normally
distributed around the central value of the Gaussian fit (± standard deviation) of $-119 \pm 2$
pptv. $\Delta HONO_{CES–CIMS}$ exhibits no significant slope with HONO (Figure S4e). While the
deployment out of Salina was operated under noticeably more humid conditions ($H_2O$ ranged
from 0.002 to 2.944%) than out of Boise ($H_2O$ ranged from 0.004 to 1.479%), we find no
significant correlation between $\Delta HONO_{CES–CIMS}$ and $H_2O$ mixing ratios (Figure 6e).



However, further laboratory studies, field measurements, and examination of this comparison
has revealed that the CIMS sensitivity to HONO is reduced when the instrument reaches
temperatures greater than 30°C (Figure S10). This sensitivity dependence on temperature
does not affect all compounds measured by the CIMS, and the sensitivity to $Cl_2$ and $HNO_3$
used for in-flight calibrations was independent of instrument temperature. The aircraft cabin
temperature was greatest during the eastern agricultural flights, when the CIMS instrument
temperatures were often 40°C and far greater than the typical 25°C instrument temperatures
in the laboratory when the CIMS HONO sensitivity was determined. As a consequence, the
reported CIMS HONO values were spuriously low, especially during the eastern fires, and
particularly later in flights when the aircraft temperatures were greatest. This intercomparison
has yielded new insights into the CIMS HONO detection sensitivity, and future work will
identify and implement appropriate corrections to this measurement.

3.3.2    Literature aircraft and ground HONO measurement comparisons
HONO measurements are notoriously difficult due to the potential for artifacts associated
with inlet surfaces as well as interferences associated with some methods (e.g., Kleffmann et
al., 2006; Xu et al., 2019). Past ground-based intercomparisons often revealed significant
discrepancies in HONO measurements. For example, six ground-based HONO measurement
techniques including a CIMS instrument were compared during the Study of Houston
Atmospheric Radical Precursors (SHARP) campaign in 2009 (Pinto et al., 2014). While three
out of six of these techniques agreed within 20%, larger deviations were found when the
other three instruments were considered and attributed to the physical separation of these
instruments. Three different techniques, including a CIMS instrument, were used to measure
HONO in the urban area of Shanghai, China (Bernard et al., 2016). The percent difference
between these measurements ranged from 27 to 46%. In 2019, six HONO measurement
techniques were again compared in a Chinese urban area, this time in Beijing, and included a
CIMS instrument as well as two broadband cavity enhanced absorption spectrometers
(BBCEAS) (Crilley et al., 2019). Percent differences up to 39% were observed during this
intercomparison and again attributed to the physical distance separating inlets coupled to high
spatial heterogeneity of HONO mixing ratios. Airborne measurements of HONO by CIMS
and CES were made during the Southeast Nexus Experiment (SENEX), and the CES
instrument was approximatively 25% higher than the CIMS instrument (Neuman et al.,
2016).


3.4 $NO_y$
3.4.1    Campaign-wide comparison
The 1Hz data comparison between the total $NO_y$ measurement by CL and $\Sigma NO_y$ is shown in
Figure 9. $\Sigma NO_y$ definition is given by Eq. 2 (see section 2.2.8). $C_1$–$C_5$ alkyl nitrates and other
minor $NO_y$ species (including $N_2O_5$, $CH_3NO_2$, and alkene hydroxy nitrates) contributed less
than 7% of the $NO_y$ budget on average (Figure 10). The overall comparison yielded a slope
(± combined instrument uncertainties) of 1.00 ± 0.25 ($R^2$ =0.98) and an intercept of –0.52 ±
0.01 ppbv (Figure 9a). The regression analysis of smoke plume-integrated $NO_y$ mixing ratios
yields a slope of 1.00 ($R^2$ = 0.99) for the whole dataset (Figure 9c). Comparison of CL $NO_y$


to $\Sigma NO_y$ in fresh (<1h since emission) and aged (>1h since emission) smoke during the
wildfires sampling period showed similar agreement (slopes of 0.98 and 1.05, respectively)
despite the chemical evolution of $NO_y$ species, highlighted by the different proportion of
those species to the $NO_y$ balance (Figure S11). Measurements used in Eq. 2 are CL $NO_x$,
CIMS HONO, CIMS $HNO_3$, HR-AMS $pNO_3$ and CIMS APNs. These measurements were
primarily used because they had better precision. Using LIF NO, CES $NO_2$ and CES HONO
as primary measurements changed the correlation slope between measured $NO_y$ and $\Sigma NO_y$ by
less than 5% (Figure S12).

Despite this correlation, two modes are apparent in the overall distribution of the absolute
difference ($\Delta NO_{yCL-Sum}$) between $\Sigma NO_y$ and the total $NO_y$ measurement (Figure 5f). The first
mode is distributed around $-0.068 \pm 0.001$ ppbv (central value of the first mode of the
Gaussian fit), while the second is distributed around an average value of $0.158 \pm 0.009$ ppbv
(central value of the second mode of the Gaussian fit). Separating the comparison into three
time periods reveals that this two-mode distribution of $\Delta NO_{yCL-Sum}$ comes from the eastern
fires sampling period as well as from the LA Basin flights whereas during the wildfires
sampling period $\Delta NO_{yCL-Sum}$ distribution is unimodal (Figure 11).

Higher $\Sigma NO_y$ compared to $NO_y$ (first mode) could be explained by (i) a lower conversion
efficiency of one or more $NO_y$ species in the CL instrument than estimated in the laboratory,
(ii) sampling loss of $pNO_3$ through the $NO_y$ inlet, and (iii) inaccuracy in one of the individual
$NO_y$ species measurement techniques. Here, we further investigated the sampling loss of
$pNO_3$ through the CL instrument $NO_y$ inlet using a multistage flow model following the
template of the Particle Loss Calculator (von der Weiden et al., 2009). The model calculates
aerodynamic losses at each stage of the $NO_y$ inlet and provides the resulting total $pNO_3$
sampling efficiency (See Section S1 and Figure SA). We find that the main aerosol sampling
loss occurs at the $NO_y$ inlet tip orifice (1.0 mm in diameter) due to the inlet orientation
(perpendicular to the aircraft flight direction). Additional loss was calculated to be negligible
once $pNO_3$ penetrated the $NO_y$ inlet, meaning that $pNO_3$ is fully volatilized into NO inside
the heated gold catalyst (See Section S1 and Figure SA). Particle sampling through the $NO_y$
inlet is highly dependent on altitude, air speed (see section S1 and Figure SB) and $pNO_3$ mass
size distribution (Figure 12a). Figure 12b shows the average modelled particle sampling
fraction through the $NO_y$ inlet, given as a ratio where a value of 1 means the total $pNO_3$ is
sampled, for each flight during FIREX-AQ. Particle sampling fraction was calculated for
three different air speeds for each flight: 40%, 65%, and 100% of the aircraft speed. An
assumed sampled air speed of 65% that of the aircraft improved the correlation between
$\Delta NO_{yCL-Sum}$ and the modelled $pNO_3$ loss in the inlet (see Section S1 and Figure SB). At that
speed, the calculated average particle sampling fraction varied between 0.36 and 0.99 for
each flight (Figure 12b). Consequently, 0–24% of the measured $NO_y$ in smoke (assuming a
sampled air speed 65% that of the aircraft) initially attributed to $pNO_3$ may result from other
reactive nitrogen species than those included in the $\Sigma NO_y$ (Figure 12b). This additional
contribution has a large uncertainty because the model may underestimate $pNO_3$ sampling
through the $NO_y$ inlet due to the large uncertainty when the losses are calculated at high air


speed (see Section S1). Further, we used bulk aerosol volume size distributions measured
with a Laser Aerosol Spectrometer (LAS; Moore et al., 2021) to derive $pNO_3$ sampling
fractions in Figure 12a as $pNO_3$ mass size distribution measurements were not available for
all flights during FIREX-AQ. At a typical FIREX-AQ sampling altitude of 5 km, the LAS
and HR-AMS size distributions can differ by about 10% (See Section S1 and Figure SC),
which adds to the uncertainty of the $pNO_3$ sampling fraction through the $NO_y$ inlet.
Correcting for particle sampling through the $NO_y$ inlet still yields an agreement between
measured $NO_y$ and $\Sigma NO_y$ that is within the combined instrument uncertainties of 25%.
On the other hand, the positive $\Delta NO_{yCL–Sum}$ mode (second mode) may indicate either an
inaccuracy in one of the individual $NO_y$ species measurement techniques or an $NO_y$ species
not measured. Further, we find that positive $\Delta NO_{yCL–Sum}$ occurred both in smoke (Figure 11d)
and in background air (Figure 11c) when sampling the eastern fires and that $\Delta NO_{yCL–Sum}$
exponentially decreased with altitude, a pattern also observed during the LA Basin flights but
not during the wildfires sampling period (Figure 13b). Note that flight altitude when
sampling the wildfires was 4.6 km on average, higher the altitude average of 0.6 and 1.1 km
during the eastern fires and the LA Basin flights, respectively. Both water vapor and $C_1$–$C_5$
alkyl nitrates (not included in $\Sigma NO_y$ thus far) were enhanced at lower altitude and may be
possible causes for the positive $\Delta NO_{yCL–Sum}$ mode. Alkyl nitrates have been shown to account
for a significant fraction of the $NO_y$ budget in past studies (e.g., Fisher et al., 2016; Hayden et
al., 2003; Horii et al., 2005). However, we find only a weak correlation between $\Delta NO_{yCL–Sum}$
and $C_1$–$C_5$ alkyl nitrates during both the wildfires ($R^2 = 0.07$) and eastern fires ($R^2 = 0.08$)
sampling periods (Figure 13c). The correlation is stronger ($R^2 = 0.44$) during the LA Basin
flights (Figure 13c). Further, we find that $C_1$–$C_5$ alkyl nitrates contributed similarly to the
$NO_y$ budget when smoke from the wildfires (1.1% on average) and the eastern fires (0.8% on
average) was sampled (Figure 10a), while the positive mode in the $\Delta NO_{yCL–Sum}$ distribution is
present in the latter period only. $H_2O$ is a known source of interference in most instruments,
and its impact on measurements is minimized when an accurate correction can be applied.
Increasing $\Delta NO_{yCL–Sum}$ is associated with increasing $H_2O$ mixing ratios in the eastern fires,
although the correlation is weak ($R^2 = 0.05$) due to the elevated scatter of the data (Figure
13a). Similar slopes and intercepts were obtained when separately comparing $NO_y$
measurements in smoke from the wildfires and eastern fires (Figures 9b and S13). The slope
of 0.81 during the LA Basin flights, may be caused by the lower precision of $\Sigma NO_y$ than that
of the CL $NO_y$ (Figure 9b).
3.4.2    Discussion and other $NO_y$ measurement comparisons

Overall, the agreement between the total $NO_y$ measured by the CL instrument and the $\Sigma NO_y$
is within instrument uncertainties. Budget closure implies that the historical definition of $NO_y$
(*i.e.*, $NO_x$ and its oxidation products, excluding reduced nitrogen species such as $NH_3$ and
HCN) is adequate even in extremely reactive conditions that foster rapid changes in $NO_y$
speciation. Reduced nitrogen species such as hydrogen cyanide (HCN) or ammonia ($NH_3$)
represent a large fraction of the total nitrogen emission from biomass burning (Roberts et al.,
2020) and have been shown to cause a small interference in CL instruments in dry air (Fahey





et al., 1985, 1986). This interference is often neglected because of either the low atmospheric
abundance of these species or sampling in humid air where such an interference is thought to
be negligible. Here, we find no evidence for a potential interference of HCN or NH$_3$, despite
their high abundance (tens of ppbv) in smoke plumes (Figure S14). Altogether, our findings
show that the NO$_y$ instrument provides an accurate and conservative measurement of total
reactive nitrogen species, although further work is needed to empirically characterize pNO$_3$
sampling through the NO$_y$ inlet.
There are a few studies that recently examined the NO$_y$ budget closure from aircraft
measurements. Calahorrano et al. (2020) presented reactive odd nitrogen partitioning during
the Western wildfire Experiment for Cloud chemistry, Aerosol absorption and Nitrogen (WE-
CAN) that sampled western American wildfires during the summer 2018. The authors found
significant (15–26%) contribution of organic N species other than APNs and alkyl nitrates to
ΣNO$_y$. However, there was no total NO$_y$ measurement during WE-CAN, and the conclusion
is based on summed individual reactive nitrogen species. The FIREX-AQ comparison of
ΣNO$_y$ to total NO$_y$ finds 2-13% of the total NO$_y$ unaccounted for, smaller than the estimate of
a 15-26% contribution from multifunctional organic nitrates from WE-CAN. While the
FIREX-AQ NO$_y$ difference suggests a smaller contribution from organic nitrates, the WE-
CAN estimate is within the uncertainty of the FIREX-AQ analysis. During the WINTER
campaign, budget closure of NO$_z$ (=NO$_y$ – NO$_x$) was demonstrated to occur within 20% for
all flights following the comparison of ΣNO$_z$ with total NO$_z$ from three different
measurement techniques, including a CL instrument (Sparks et al., 2019). A recent ground-
based study in New York State in the US found that the sum of the individual reactive odd
nitrogen species accounted for 95% of the total NO$_y$, well within measurement uncertainties
(Ninneman et al., 2021). These recent studies contrast with somewhat older literature that
often reported a significant shortfall in the NO$_y$ balance, where measured NO$_y$ was higher
than ΣNO$_y$ (Hayden et al., 2003; Horii et al., 2005; Williams et al., 1997; Zhang et al., 2008).
This shortfall has often been attributed to unmeasured organic N species and more
specifically alkyl nitrates (Day et al., 2003; Horii et al., 2005). During FIREX-AQ, C$_1$–C$_5$
alkyl nitrates accounted for less than 7% on average of the NO$_y$ budget (Figure 10),
consistent with findings from other regions in the US (Benedict et al., 2018; Russo et al.,
2010). However, FIREX-AQ did not include a measurement of total alkyl nitrates.
3.5 CO
3.5.1    Campaign-wide comparison
The 1Hz data comparison between the ICOS and the TDLAS instruments is shown in Figure
14. The overall comparison yielded a slope (± combined instrument uncertainties) of 0.98 ±
0.03 (R$^2$ = 0.99) and an intercept of –1.06 ± 0.01 ppbv (Figure 14a). The regression analysis
of smoke plume-integrated CO mixing ratios yields a slope of 0.99 (R$^2$ = 1) for the whole
dataset (Figure 14c). A histogram of the absolute difference between CO measurements
(ΔCO$_{ICOS–TDLAS}$) is shown in Figure 5g. 90% of the values were between –6.05 and 2.35 ppbv,
and the whole dataset is normally distributed around the central value of the Gaussian fit of –
2.87 ± 0.02 ppbv. This is indicative of an offset between the two CO instruments, with the





TDLAS systematically higher than the ICOS instrument. This average 2.87 ppbv offset was
consistent throughout the campaign regardless of the type of fires that were sampled.
Therefore, it cannot explain the significantly lower agreement of the instruments during the
eastern fires compared to the wildfires sampling period (Figures 14b and S12). During the
first period, the overall slope was 0.99 and ranged from 0.97 to 1.02 (average of 0.99) for
individual flights, well within the combined instrument uncertainties of 3% (Figures 14b and
S12). However, all individual flight measurements during the eastern fires sampling period
exhibit slopes reduced by about 10% (range = 0.86–0.91 with an average of 0.89) and largely
positive intercepts (range 6.75–19.04 with an average of 11.51) (Figure S15). As observed for
other species, the second period proved to be a more challenging environment for CO
measurements. This may be attributed to a spectral issue with one or the other of these two
instruments, although we could not identify the source of the discrepancy. $\Delta CO_{ICOS–TDLAS}$
exhibit no significant slope with CO (Figure S4g) and $H_2O$ (Figure 6g) mixing ratios.

3.5.2    Literature aircraft CO measurement comparisons

Overall, the comparison between the two CO instruments shows an agreement well within
stated uncertainties. We find that the agreement between the two CO instruments used during
FIREX-AQ is well in line with past intercomparisons. During the GTE-CITE experiment, the
comparison of a TDLAS technique with two grab sample/gas chromatograph methods for
detection of CO showed agreement across the instruments – within the combined instrument
uncertainties and strong correlations ($R^2 = 0.85$–$0.98$) for CO ranging from 60 to 140 ppbv
(Hoell et al., 1987). During the North Atlantic Regional Experiment (NARE 97) CO was
measured by TDLAS and vacuum ultra-violet fluorescence with agreement to within 11%
and systematic offsets of less than 1ppbv (Holloway et al., 2000). CO was also more recently
measured by TDLAS and vacuum ultra-violet fluorescence during the side-by-side
comparison of instruments aboard two aircraft during the DC3 experiment. There, CO
measurements agreed within 5% during flight periods typically ranging from 20 to 30
minutes (Pollack et al., 2016).

**4   Conclusion**
In this study, we compare airborne measurements of NO, $NO_2$, HONO, $NO_y$ and CO
conducted during the FIREX-AQ campaign in the summer 2019. This dataset offers the
opportunity to assess the accuracy of a large suite of detection techniques in a challenging
environment where species mixing ratios increased by tens of ppbv in seconds between
background air and fire smoke. For NO, $NO_2$ (CES and LIF), $NO_y$ and CO, correlations agree
better than the combined instrument uncertainties, indicating that the stated individual
uncertainties are conservative estimates. For $NO_2$ (CL) and HONO, the percent difference
between measurements is higher than the combined instrument uncertainties, indicating
potential interferences or calibration inaccuracies that are not identified at this time. Based on
the analysis above, we make the following recommendations, which are specific to the
FIREX-AQ campaign.

1) Comparison of NO measurements by LIF and CL showed an overall agreement well
within instrument uncertainties. Flight-to-flight agreement was generally more variable





during the eastern fires sampling period than during the wildfires sampling period, which was
attributed to the heterogeneous nature of smoke plumes combined with the physical
separation of inlets. Both measurements are considered reliable for FIREX-AQ, although the
LIF instrument has better 1Hz precision (1 pptv) than the CL instrument (6 pptv), and the CL
instrument exhibited slower time response.
2) Comparison of $NO_2$ measurements by LIF and CES showed an overall agreement well
within the stated instrument uncertainties. However, $NO_2$ measured by CL is on average 10%
higher than that measured by the other two techniques. The agreement worsens for all
instruments when comparing $NO_2$ measured during the eastern fires sampling period, likely
for similar reasons as indicated for the NO measurements.
3) The CES and CIMS HONO measurements were highly correlated in each fire plume
transect, but the correlation slope of CES vs. CIMS for all 1 Hz data from the entire
campaign was 1.8. The HONO measured by CIMS was on average 74% of that measured by
CES during the wildfires sampling period, and on average 40% of CES during the eastern
fires sampling period. The higher precision data from the CIMS are most useful for analysis
of HONO when mixing ratios are lower. The redundancy of HONO measurements during
FIREX-AQ led to the discovery that the CIMS sensitivity to HONO was reduced in a high
temperature environment. This intercomparison has initiated further studies of the CIMS
sensitivity to HONO and other compounds.
4) Closure of the $NO_y$ budget between the total $NO_y$ measurement by CL and $\Sigma NO_y$ was
achieved for all flights and correlation slopes were usually much better than the combined
instrument uncertainties of 25%. $NO_x$, $HNO_3$, HONO, APNs and $pNO_3$ are the main
contributors to the $NO_y$ budget, with the other reactive N species contributing less than 10%
on average. We find that the modelled $pNO_3$ sampling fraction through the $NO_y$ inlet is
highly dependent on altitude, air speed and $pNO_3$ mass size distribution, and varied on
average between 0.36 and 0.99 during FIREX-AQ. Therefore, approximately 0–24% on
average of the total measured $NO_y$ by CL may be unaccounted for and possibly explained by
other species such as multifunctional organic nitrates. The reason for the secondary positive
mode of 0.4 ppbv in the $\Delta NO_{yCL-Sum}$ distribution in the eastern fires and LA Basin flights
could not be clearly identified. Potential explanations include the contribution of gas-phase
organic nitrates, not included in the $\Sigma NO_y$, and/or a water vapor interference in one or more
instruments. Regardless, we conclude that the total $NO_y$ measurement by CL provides a
robust quantification of the reactive nitrogen species in background air as well as in smoke
plumes, and that the total $NO_y$ measurement is not sensitive to interference from reduced
nitrogen species in fire plumes. Further laboratory and field work will be needed to fully
characterize $pNO_3$ sampling through the $NO_y$ inlet.
5) Comparison of CO measurements by TDLAS and ICOS showed an agreement well within
the combined instrument uncertainties. An offset of ~2 ppbv between the two instruments
was identified but has little impact on the correlation. There was a clear difference in the



agreement between the wildfires sampling period and the eastern fires sampling period,
where the correlation slopes were about 10% lower.
6) Integrating data across smoke plume transects generally improved the correlation between
independent measurements and may be necessary for fire-science related analyses, especially
for smaller plumes with greater spatial heterogeneity compared to the distance between the
sampling locations on a large aircraft.
**Data availability**
All data used in this manuscript are archived online and available at https://www-
air.larc.nasa.gov/cgi-bin/ArcView/firexaq.
**Author contribution**
I.B. and T.B.R. designed research. All authors performed FIREX-AQ measurements. P.C.-J.,
H. G., and J.L.J performed the flow modelling analysis. All authors analyzed data. I.B., J.P.,
J.A.N., and S.S.B. wrote the original draft and all authors edited and revised the paper.
**Competing interests**
The authors declare they have no conflict of interest.

**Acknowledgments**
We would like to thank the NOAA/NASA FIREX-AQ science and aircraft operation teams.
We acknowledge A. Whistaler, F. Piel and L. Tomsche for providing the $NH_3$ measurements
from FIREX-AQ. We thank A. Middlebrook for helpful discussion regarding $pNO_3$ sampling
in the $NO_y$ inlet and AMS performance. IB, JP, JAN, SSB, MMC, JBG, GIG, AL, PSR,
MAR, AWR, RAW and CCW were supported by the NOAA Cooperative Agreement with
CIRES, NA17OAR4320101. VS acknowledges NOAA grant NA16OAR4310100. JMS and
TFH acknowledge support from the NASA Tropospheric Composition Program and NOAA
Climate Program Office's Atmospheric Chemistry, Carbon Cycle and Climate (AC4)
program (NA17OAR4310004). DP, BAN, HG, PCJ and JLJ were supported by NASA grant
80NSSC18K0630. LX, KTV, HA and POW acknowledge NASA grant 80NSSC18K0660.

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



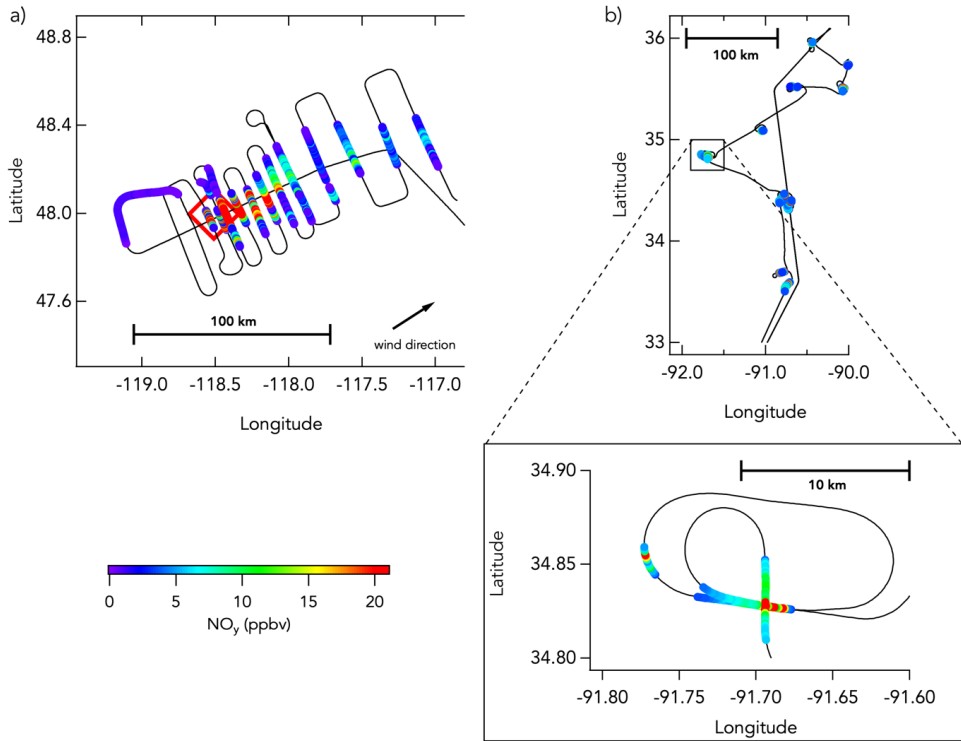

**Figure 1** Examples DC-8 flight tracks from western wildfires and eastern agricultural fires. Panel a) shows the DC-8 flight track (black line) during the sampling of the Williams Flat fire (03/08/2019) smoke plume, colored by $NO_y$ mixing ratios (only data in smoke are colored here). Panel b) shows the DC-8 flight track during the sampling of multiple agricultural burns (21/08/2019), also colored by $NO_y$ mixing ratios (only data in smoke are colored here).

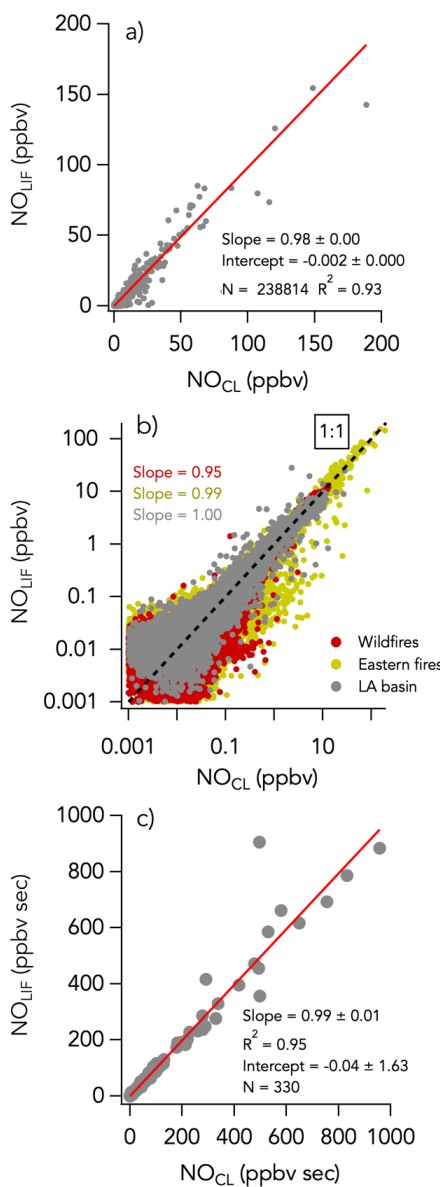

**Figure 2** NO measurements by LIF versus CL with a) 1 s data on a linear scale, b) 1 s data on a log scale, and c) integrals of 330 crosswind smoke plume transects. N is the number of independent 1 s observations or smoke plume transects that are compared. In panel b, the three sampling periods are shown in different colors with the wildfires sampling period in red, the eastern fires sampling period in mustard, and the Los Angeles (LA) Basin flights in grey. The red lines indicate the fit of the data. The dotted black line is the 1:1 line.

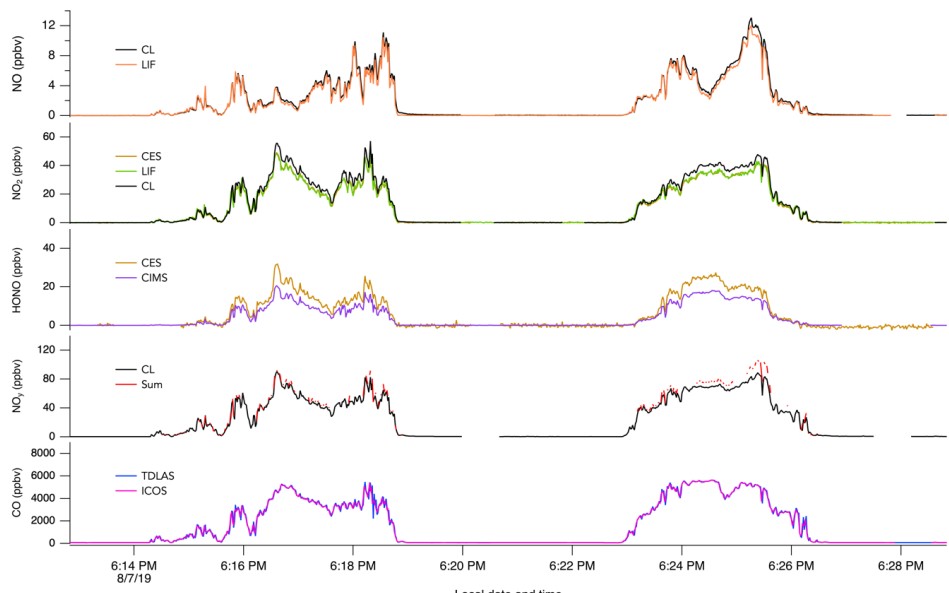

**Figure 3** 1 s measurements of a) NO, b) $NO_2$, c) HONO, d) $NO_y$, and e) CO during two crosswind plume transects of smoke from the Williams Flat fire on 07/08/2019. The plume transects were chosen due to the significant enhancement of all species at that time. Note that in panel b) the $NO_2$ trace from the CES instrument is hidden behind the $NO_2$ trace from the LIF instrument.



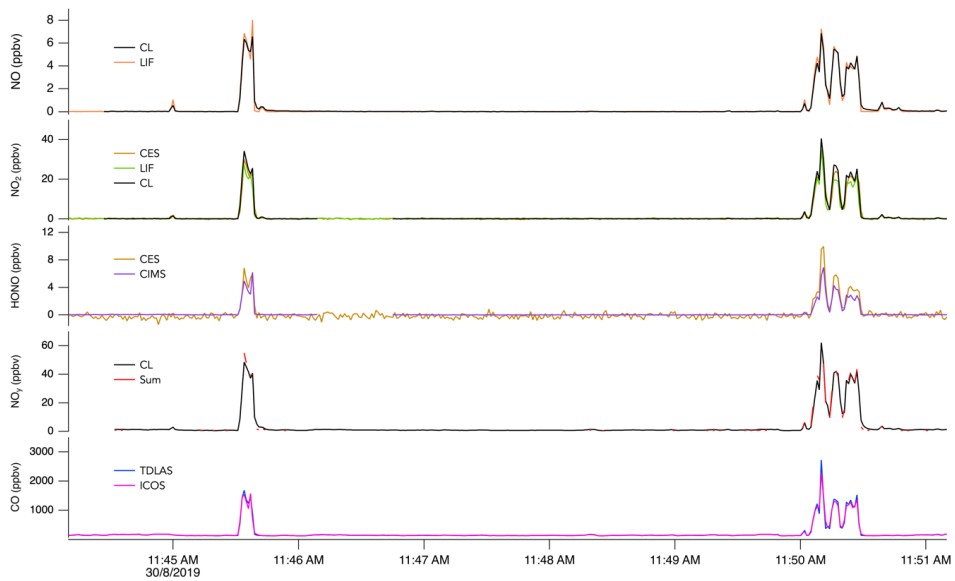

**Figure 4** 1 s measurements of a) NO, b) $NO_2$, c) HONO, d) $NO_y$, and e) CO during crosswind plume transects of smoke from crop burning in southeastern US on 30/08/2019.

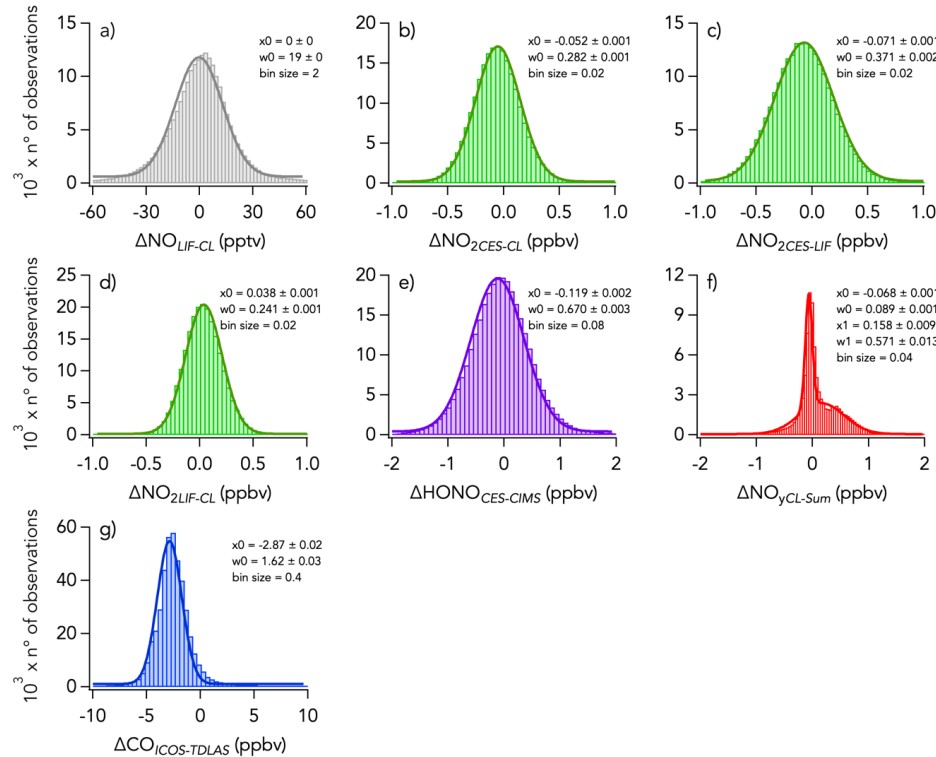

**Figure 5** Histograms of the absolute difference of 1 s measurements of a) NO, b)–d) $NO_2$, e) HONO, f) $NO_y$, g) CO for the entire campaign. Parameters of the gaussian fit to the histogram is indicated in each panel with x0 and w0 being the central value and the width of the fit, respectively. Note that in panel f) a double gaussian was fitted to the histogram and that the parameters for the second mode are given by x1 and w1.

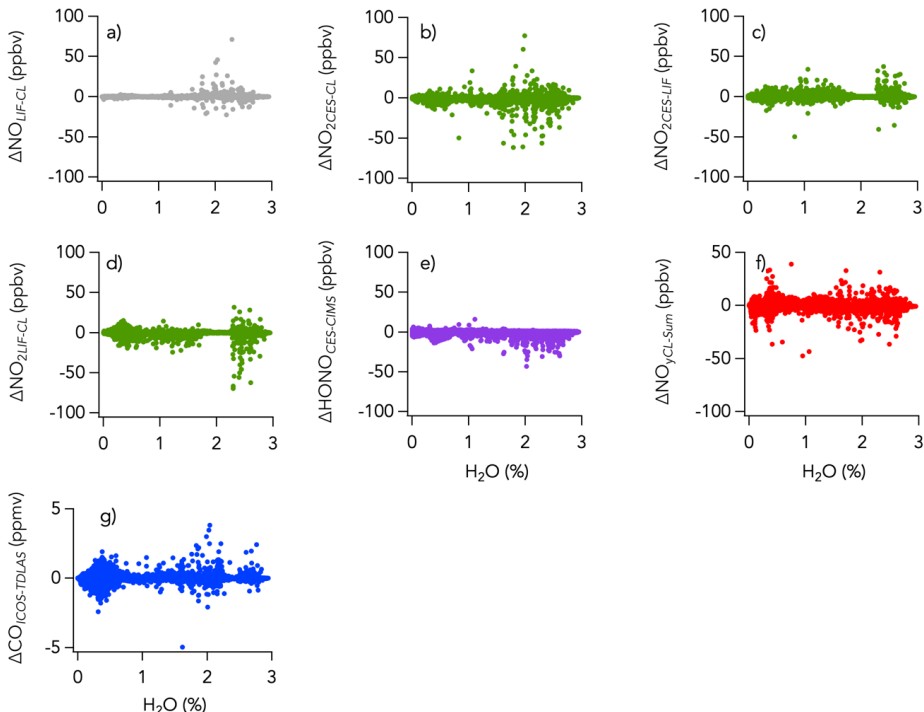

**Figure 6** Measurement difference (1 s data) of a) NO, b)–d) $NO_2$, e) HONO, f) $NO_y$, g) CO as a function of water vapor for the entire campaign.



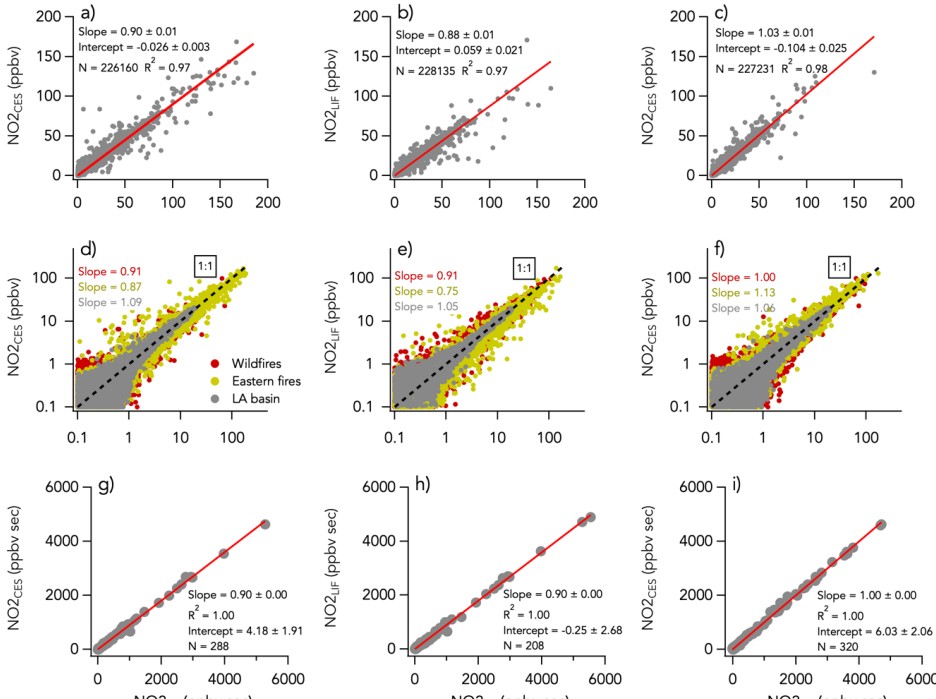

**Figure 7** NO₂ measurements by LIF, CES and CL with a)–c) 1 s data on a linear scale, d)–f) 1 s data on a log scale, and g)–i) integrals of 208–320 crosswind smoke plume transects. N is the number of independent 1 s observations or smoke plume transects that are compared. In the panels d)–f), the three sampling periods are shown in different colors with the wildfires sampling period in red, the eastern fires sampling period in mustard, and the Los Angeles (LA) Basin flights in grey. The red lines indicate the fit of the data. The dotted black lines are the 1:1 line.

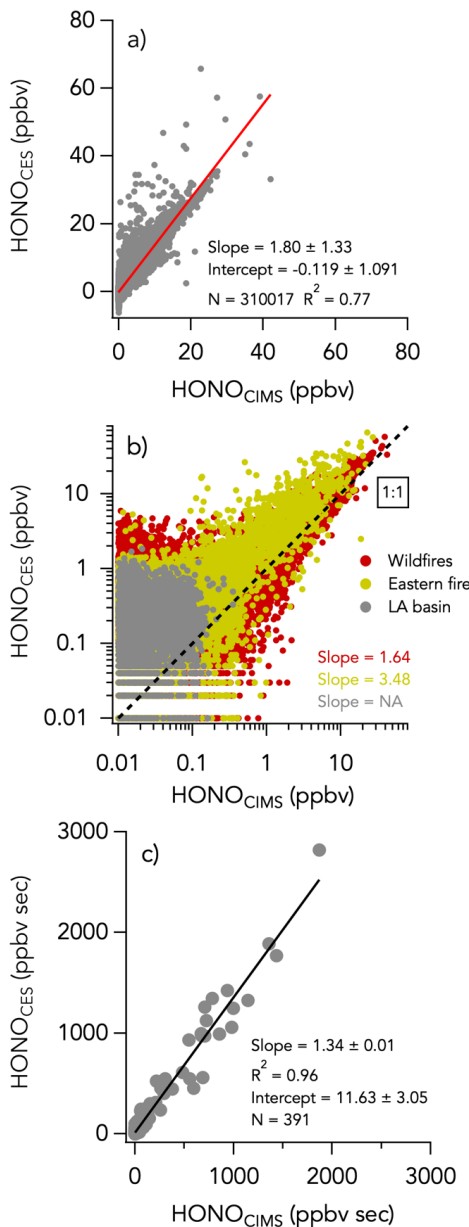

**Figure 8** Same as Figure 2 but comparing HONO measurements by CES and CIMS. No slope is given for the Los Angeles (LA) flights in panel as most of the HONO signal at that time was below the instruments' detection limits.

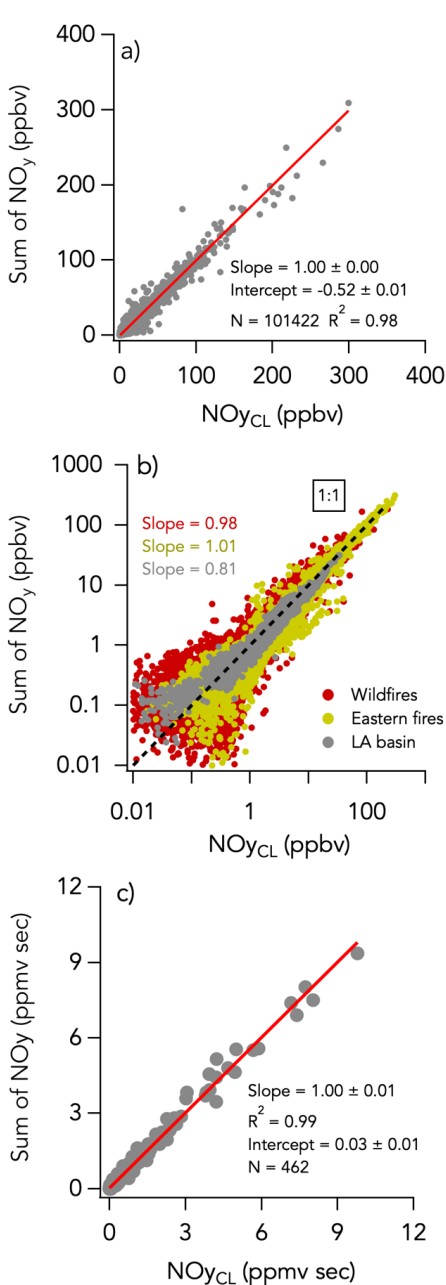

**Figure 9** Same as Figure 2 but comparing the sum of individually measured NO$_y$ species (= NO$_x$ + HONO + HNO$_3$ + APNs + pNO$_3$) with the total NO$_y$ measurement by CL.

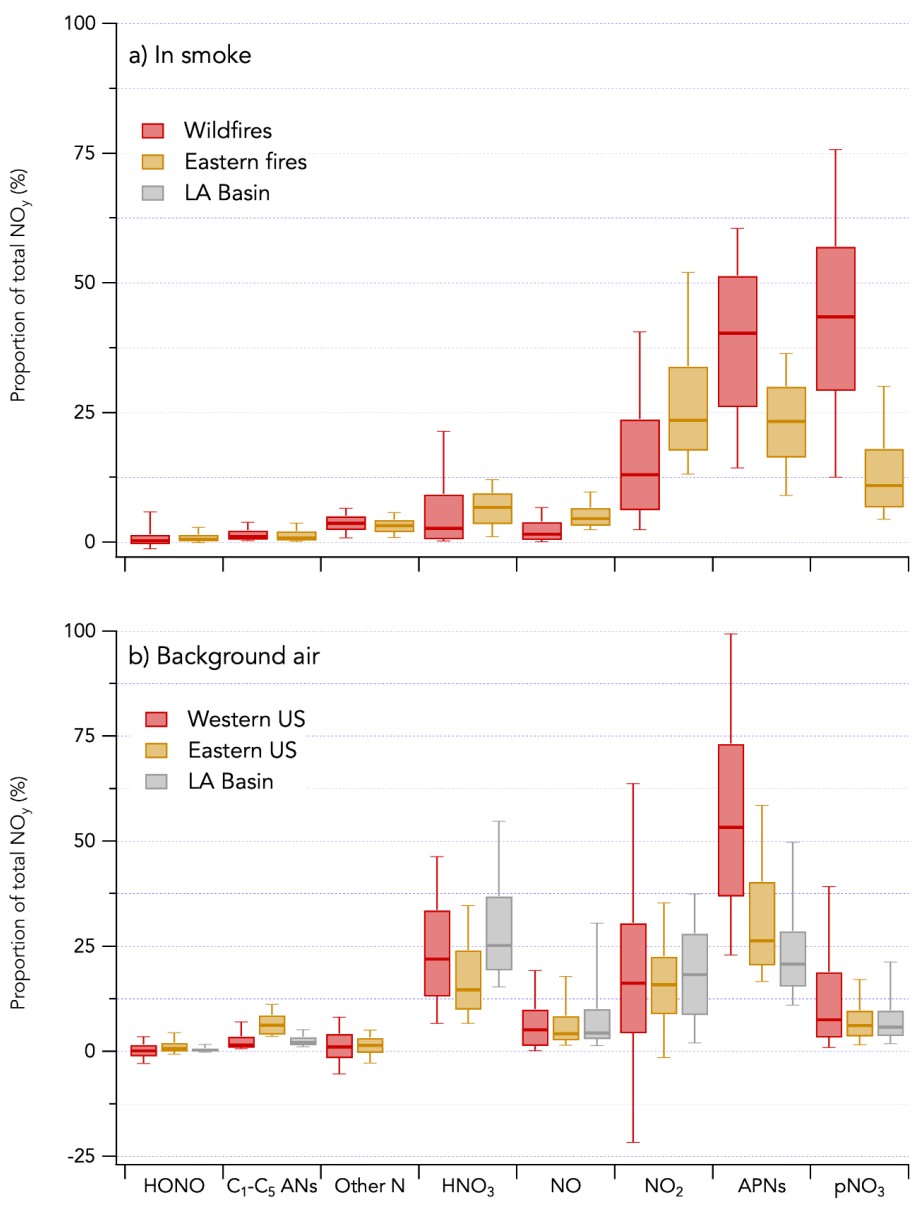

**Figure 10** Contribution of individually measured reactive odd nitrogen species to the total $NO_y$ budget during FIREX-AQ. The campaign is separated in three periods (wildfires sampling period in red, eastern fires sampling period in yellow, and Los Angeles (LA) Basin flights in grey). The panel a) show the $NO_y$ budget in smoke plumes, while the panel b) shows that in background air. $C_1$–$C_5$ alkyl nitrates are referred to as $C_1$–$C_5$ ANs. Other nitrogen species include $N_2O_5$, $CH_3NO_2$, and alkene hydroxy nitrates.

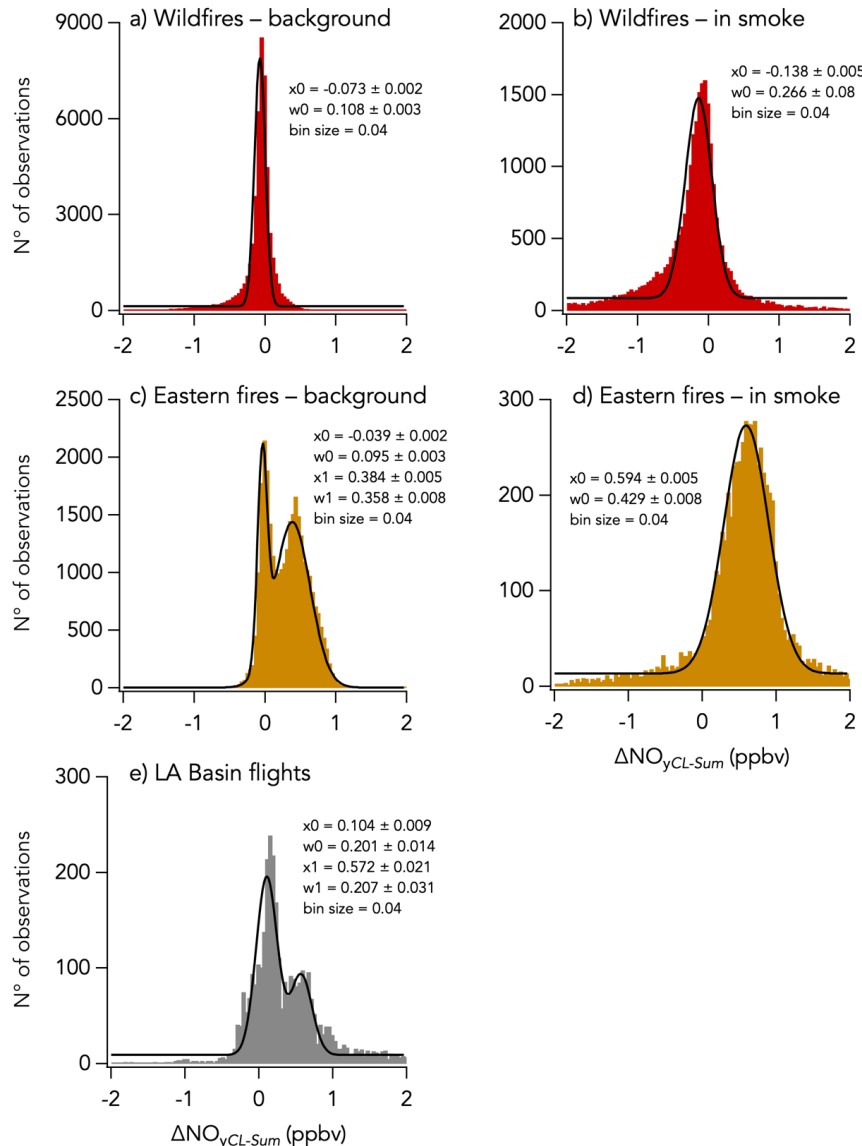

**Figure 11** Histograms of $\Delta NO_{yCL\text{-}Sum}$ for three sampling periods during FIREX-AQ with the wildfires sampling period in red, the eastern fires sampling period in yellow, and the Los Angeles (LA) Basin flights in grey. Further separation was made between in smoke measurements (panels b and d) and background air measurements (panels a, c, and e). Parameters of the gaussian fit to the histogram is indicated in each panel with x0 and w0 being the central value and the width of the fit, respectively. Note that in the panels c) and e) a double gaussian was fitted to the histogram and that the parameters for the second mode are given by x1 and w1.

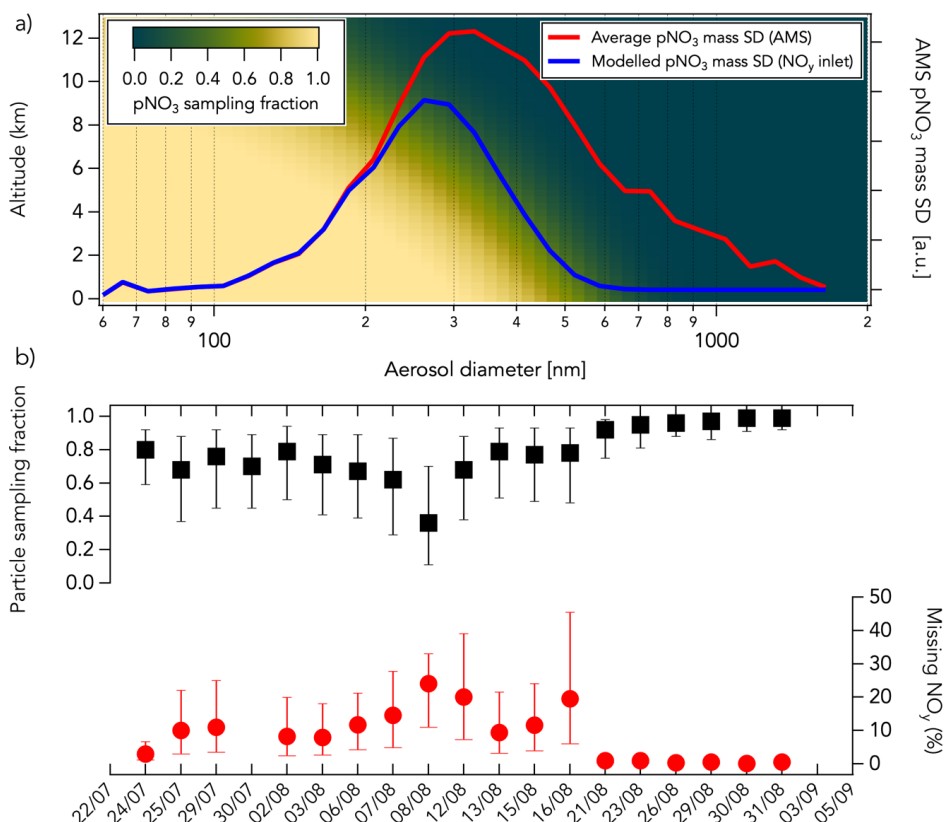

**Figure 12** Panel a): The modelled pNO₃ sampling fraction through the NOy inlet as a function of altitude and pNO₃ mass size distribution (SD) is shown with a gradient of color from green (low sampling fraction) to yellow (high sampling fraction). The average pNO₃ mass size distribution measured in the Williams Flat fire smoke on 07/08/2019 by HR-AMS is shown in red. The modelled pNO₃ size distribution sampled in the NOy inlet assuming an altitude of 5km and a sampled air speed 65% that of the aircraft is shown in blue. In this example case, the sampled pNO₃ mass fraction is ~50%. Panel b): The average modelled particle sampling fraction in the NOy inlet (in black) and the corresponding percentage of measured NOy that may be unaccounted for (in red) are shown for each flight assuming a sampled air speed of 40% (bottom bars), 65% (markers) and 100% (top bars) that of the aircraft speed. The sampling fractions were calculated using bulk aerosol volume distributions measured by a Laser Aerosol Spectrometer (see Section S1 and Figure SC). The missing NOy corresponds here to the percentage of measured NOy that pNO₃ not sampled through the NOy inlet represents. Data shown in the panel b) are from air in smoke only.

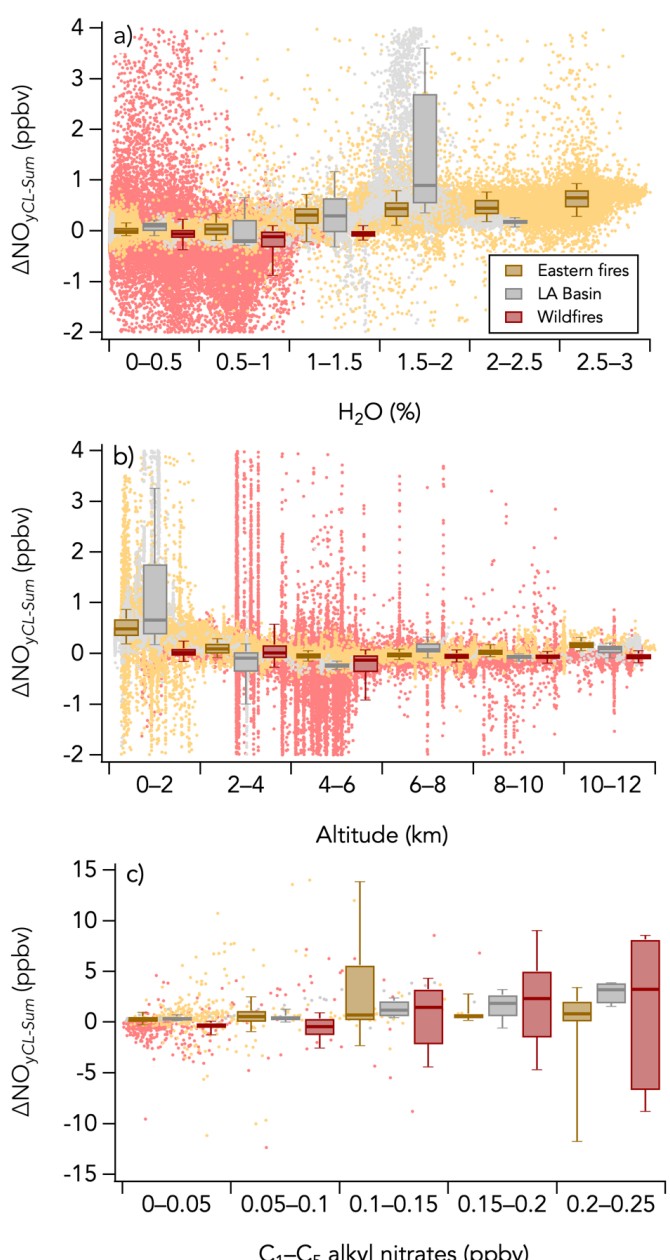

**Figure 13** Scatterplots of a) $\Delta NO_{yCL\text{-}Sum}$ vs $H_2O$, b) $\Delta NO_{yCL\text{-}Sum}$ vs altitude and c) $\Delta NO_{yCL\text{-}Sum}$ vs $C_1$–$C_5$ alkyl nitrates measured by the iWAS instrument for three sampling periods during FIREX-AQ (wildfires sampling period in red, eastern fires sampling period in yellow, and Los Angeles (LA) Basin flights in grey). The box and whisker plots show the 10th, 25th, 50th, 75th, and 90th percentiles of $\Delta NO_{yCL\text{-}Sum}$ distributions in each bin. The dots are the 1Hz data in panels a) and b), and 1Hz data averaged to match the iWAS sampling time in panel c).

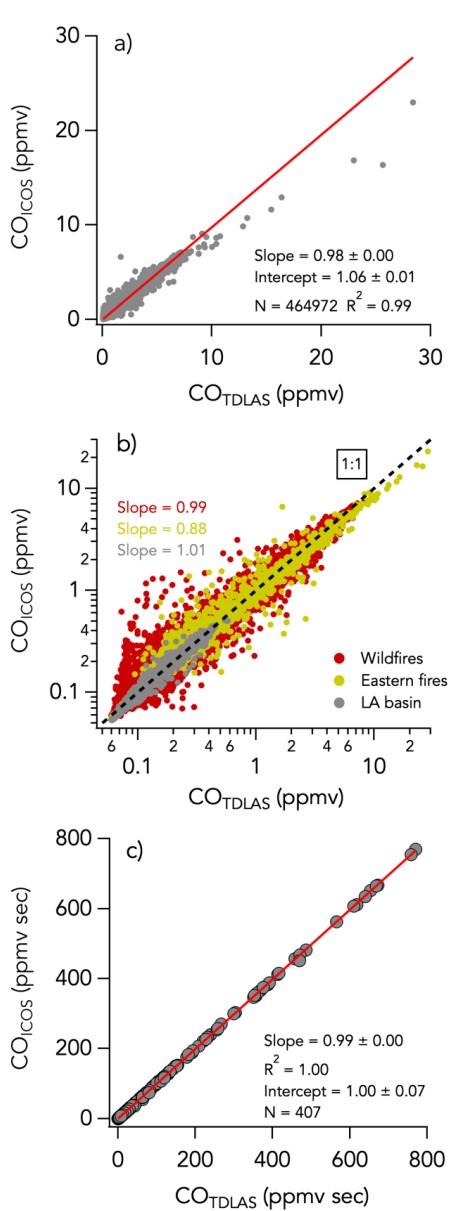

**Figure 14** Same as Figure 2 but comparing CO measurements by TDLAS and ICOS.