# Peer review of "Comparison of airborne measurements of NO, NO2, HONO, NOy and CO during FIREX-AQ"

_Atmospheric Measurement Techniques, 2021_

## Author Comment (AC1)

We thank the reviewers for their time and constructive comments that have improved our manuscript. Below we include specific responses to the reviewer's comments. The reviewer's comments are in black. Authors' responses are in blue, quotes from the manuscript are in *italic*, and changes to the text are shown in red.
* * *
Thank you for giving me the opportunity to review this manuscript. I hope my input will be useful to the authors.

I appreciate this paper very much. The paper describes results from an intercomparison of different measurement techniques for what many would call "basic" photochemical tracers but the validity and precision of these tracer measurements are in fact critical for the success of any atmospheric chemistry mission.

This paper is a timely and very relevant contribution at a time during which measurement capabilities are expanding rapidly. It is well structured and clearly written using precise language. The scientific methods are sound. The content is entirely appropriate for publication in AMT. The reference list is appropriate. I recommend accepting the manuscript for publication after considering the following suggestions.

HONO:

I have a bit of a hard time wrapping my mind around the HONO comparison numbers and illustrations.

1. How can the distribution of the factional errors be so tight around a value of 2 (Figure S1)? When plugging into the equation CES/CIMS values mentioned at various places in the manuscript (1.36, 1.8, 2.48 and 3.9) I calculate values for FE between 0.3 and 1.2. This should center the gaussian somewhere in the middle of that range and show a much broader distribution.

2. Figure S4e shows that the CES measures anywhere from zero to 25 ppb while the CIMS measures between 0 and 3 ppb. How does this reconcile with the gaussian distribution shown in Figure 5 and the mean difference plots in figure S9?

This might require a bit more explanation than what is currently presented in the text.

Response:

1. In Figure S1, only data from smoke is presented (as indicated in the caption) whereas the slopes in the main manuscript are for all 1Hz data (i.e., smoke + background). Even so, there is very little HONO left in most of the data collected in smoke: about 90% of HONO data in smoke is below 1 ppbv, which is about the precision of the CES instrument. The fractional error (FE) expression is as follows:

FE = (CES HONO – CIMS HONO) / Average HONO

Since Average HONO = (CIMS HONO + CES HONO) / 2 we get

FE = 2*(CES HONO – CIMS HONO) / (CIMS HONO + CES HONO)

When HONO mixing ratios are close to 0, CES HONO >> CIMS HONO because of the lower precision of the CES. Hence, FE becomes tightly distributed around a value of 2.

We added a sentence in the caption of Figure S1 (now Figure S3) to emphasize this point.

"*The tight distribution of FE-HONO$_{CES-CIMS}$ around a value of 2 is due to the lower precision of the CES instrument when HONO mixing ratios were close to 0 (~90% of the data in smoke).*"

2. Few data points actually contributed to the pattern identified by the reviewer. We realized that this figure may be misleading and we replaced it with a new figure (now Figure S6) showing the same plots but color coded by data density.

[Figure]

**Figure S6** Measurement differences (1Hz data) of a) NO, b)–d) NO$_2$, e) HONO, f) NO$_y$, g) CO as a function of the species mixing ratios for the entire campaign. The color bar indicates the number of individual data points per bin of mixing ratios (bin size is 2.5×2.5 ppbv).

While I can appreciate that the IMR temperature may have an influence on sensitivity I'd be curious to know why this would only affect HONO and not also other analytes. Other readers might be left wondering about this.

Response: Other analytes do show temperature dependences to varying degrees. See response and reference below, where this subject is discussed in detail.

Was there an attempt made to correct the HONO values for IMR temperature, and how does the comparison look like then?

Response: A correction has indeed been derived for the CIMS HONO value based on the temperature. The details of this correction are beyond the scope of this manuscript and is the object of a follow-up paper focusing specifically on this issue that is currently under review in *AMTD* (Robinson et al., 2022). We added the new reference to the text lines 743-745.

"*This intercomparison has yielded new insights into the CIMS HONO detection sensitivity, and future work will identify and implement appropriate corrections to this measurement (Robinson et al. 2022).*"

NOy:

For lack of a better word, I find the assessment of the NOy measurements somewhat sugarcoated. In my view there are too many uncertainties to make these measurements ultimately useful, at least for fire smoke research. The facts I gather are the following:

1. Particulate nitrate makes up the largest fraction of total NOy in western wildfire smoke and a significant fraction in the eastern fires.
2. The sampling efficiency of particulates is highly dependent on airspeed but the real airspeed at the inlet tip (and the dependency on type of aircraft, banking and attack angle, or install location on the aircraft) is unknown.
3. Particulates used in the model described are assumed to be ammonium nitrate. The exact composition of the nitrates contained in fire smoke particulates is not known. There could be a significant fraction of organics, in particular nitroaromatics, but their volatilization behavior and conversion efficiency in the gold converter is unknown.
4. In the best case of sampling efficiency, 25% of the nitrate could be unaccounted for. Looking at the graph in Figure 10a, that fraction could be more than 50% in the worst case.

I'd agree with the authors if you call this a devil's advocate assessment, but at the end of the day my impression is that NOy measurements and "oxidized nitrogen closure" calculations based on these measurements or their use as photochemical clocks, etc. still need to be taken with a (fairly large) grain of salt. Just like they had to in the past.

We appreciate the reviewer's summary and agree with it to some extent.

1) Particulate nitrate makes up the largest fraction of total $NO_y$ in western wildfire aged smoke and a significant faction in western wildfire fresh smoke and in the eastern fires based on the speciated measurements.
2) The impact of air speed on the CL sampling efficiency of particles has been examined in depth in the Supplementary Information (see Section S1 - Estimation of particle losses and sensitivity to air speed). We show in Figure SB that correcting for the CL inlet transmission of particles significantly improves the correlation between $\Sigma NO_y$ and measured total NOy (without biasing the residuals), meaning that the uncertainty associated with the airspeed at the inlet tip is significantly smaller than the full range of uncertainties shown in Figure 12.

3) It is explicitly stated in section S1 that the volatility that matters for the $NO_y$ converter is the bulk aerosol volatility, not the $NH_4NO_3$ volatility. During FIREX-AQ, $pNO_3$ volatility was on par/slightly lower than that of organic aerosols but significantly higher than that of ammonium sulfate, based on in-situ measurements (Pagonis et al., personal communication). So again, this shows that using the work from Clarke et al. (1991) as done in the section S1 is justified and that indeed all non-refractory material excluding ammonium sulfate but including all organic N compounds are likely volatilized in the CL inlet. The conversion efficiency of organic $pNO_3$ and in particular of nitro-aromatics in the CL gold converter has not been characterized so far, although a previous study reports overall excellent conversion of several nitro-compounds (including nitro-aromatics) in a similar CL inlet with a gold converter (Bradshaw et al., 1998).
4) As noted in 2) the worst case called out by the reviewer is unlikely given the sensitivity test performed on sampled air speed in section S1.

What would a comparison of plume dilution calculated using CO versus a similar calculation using NOy look like?

Response: We appreciate the suggestion from the reviewer to compare the $NO_y$ to CO normalized excess mixing ratios (NEMR). A figure is shown below but not included in the manuscript. Because of changing fire conditions during the course of a day, smoke of different nominal ages may have different initial $NO_x$ to CO ratios. We do not interpret the changes in nominally conserved tracers as indicative of either aging or sampling artifacts as a result. We have added a sentence to the paper lines 884-887 to this effect:

"*The $NO_y$ to CO ratio was approximately conserved with smoke age, but showed both increasing and decreasing trends with different fires, likely as a result of variability in the $NO_x$ to CO emission ratio during the course of a day with changing fire conditions.*"

[Figure]

Minor suggestions:

Section 2 intro: Maybe a figure with a plumbing diagram of the manifolds described in the manuscript could be helpful

Response: We thank the reviewer for this suggestion. Such a figure is already included in the FIREX_AQ overview paper (Warneke et al., 2022). We added a citation for this article in the manuscript line 65.

"*The focus of the joint National Oceanic and Atmospheric Administration (NOAA) / National Aeronautics and Space Administration (NASA) Fire Influence on Regional to Global Environments and Air Quality (FIREX-AQ) airborne campaign was to provide comprehensive observations to investigate the impact of summer time wildfires, prescribed fires and agricultural burns on air quality and climate across the conterminous US (Warneke et al., 2022).*"

Section 2.2.1.: What was the conversion efficiency of the photolysis converter?

Response: The conversion efficiency is $40 \pm 1$ %. We added this information lines 171-173.

"*In the $NO_2$ channel, $NO_2$ is photolyzed to NO with a $40 \pm 1$ % conversion efficiency using two ultraviolet (UV) LEDs (Hamamatsu, model L11921) at 385 nm in a 45 cm long quartz cell (inner diameter of 1.2 cm) pressure-controlled at $209.8 \pm 0.3$ Torr.*"

Line 134: Suggest replacing "minimal" with "least possible"

Response: Done.

Line 169: NOy is missing in the list

Response: Fixed.

Section 2.2.3.: What is the inlet material? What was is shared with?

Response: The inlet tube is Silcosteel (Restek) coated with FluoroPel. It was shared with another four instruments during FIREX-AQ, two instruments from NASA Goddard (ISAF HCHO and ROZE O3) and two instruments from NOAA (SO2 and NO). We added a clarification to the text lines 240-242.

"*The inlet tube is a 45 cm length of 0.94 cm inner diameter Silcosteel (Restek) coated with FluoroPel (Cazorla et al., 2015). The CANOE instrument pulled its 750 sccm sample flow from a shared manifold (with another four instruments) at the instrument rack.*"

Line 305: How can the addition of 1% flow of saturated nitrogen stabilize the I- / I-*H20 clusters to such a precise ratio?

Response: We vary the water addition by controlling between 0 and 50 sccm of $N_2$ saturated with water, and adjusting the flow to maintain a constant reagent ion cluster ratio. This compensates for the changes in ambient humidity. The ambient airflow into the instrument is ~1 slm, with specific humidity that goes up to about 2%. Since the water addition has up to ~50x greater mixing ratio than the ambient sample, the additional water flow that is about 50x smaller than the ambient sample is sufficient to compensate for changes in ambient humidity.

Line 401: should be CH3NO2

Response: Fixed.

Line 511: If this uncertainty is based only on the spread of the trajectory ensemble, does this mean that there is additional uncertainty arising from the plume rise time, calculated using a fixed vertical transport speed?

Response: The age uncertainty provided in the data files includes multiple factors:
- Spread of trajectory ensemble (as mentioned by the reviewer)
- Uncertainty in updraft speed ($\pm 50\%$)
- For large fires, ages are calculated from several possible emission locations within the fire
- Uncertainty due to upwind trajectories not passing directly over the source fire (due to model wind errors)
- Wind speed uncertainty: since met field wind speeds don't exactly match observed wind speeds, an additional age estimate is derived by rescaling the trajectory age based on the ratio of model/observed wind speed. The uncertainty due to this effect is taken to be half of the difference between the unscaled and rescaled ages.

All of these uncertainties are added together in quadrature (i.e. assuming independent) to get the provided age uncertainty. This methodology will be described in full detail in a paper that is still in progress.
We added this precision to the text lines 546-549

"*The median uncertainty in smoke age is about 27%, as determined by the sum in quadrature of the spread among the ensemble of estimates, the uncertainties in the updraft speed, the fire location and the wind speed, and uncertainties in the model.*"

Line 575: I am not certain what the logic is behind putting some figures into the supplement and others into the main manuscript. Maybe this could be revisited.

Response: We appreciate the suggestion but have chosen main text figures in order to keep the manuscript as short as possible while placing the most important figures in the text.

Line 633: Could the positive artifact in the CL instrument be caused by thermal decomposition of peroxynitrate species inside the photolytic converter cell (which might be warmed by the heat output of the LEDs?)

Response: We thank the reviewer for this comment, but we do not believe peroxynitrates (i.e., $HO_2NO_2$ and $RO_2NO_2$) thermal dissociation to be the source of the artifact. The comparison did not depend on altitude or outside air temperature, suggesting that the artifact did not result from thermally labile species. The following sentence has been added to lines 681-682:

"*Further, $\Delta NO_{2CES-CL}$ or $\Delta NO_{2LIF-CL}$ did not depend on altitude or outside temperature, which also suggests little influence from thermally labile species.*"

Line 699: Suggest starting the paragraph with "The interpretation of literature…."

Response: We thank the reviewer for the suggestion. We modified the paragraph line 748.

Line 733: If there is an obvious problem with the CIMS HONO measurements, why are these being used here?

Response: The CIMS HONO measurements were used because they have much better precision than the CES HONO. Precision matters more than accuracy for plots like Figure 5, that are more weighted by background data than by smoke data. Using CES HONO rather than CIMS HONO only affects the slope of the correlation between measured $NO_y$ and $\Sigma NO_y$ by 6%. We clarified this aspect lines 370-372.

"*Using LIF NO, CES $NO_2$ and CES HONO as primary measurements changed the correlation slope between $\Sigma NO_y$ and measured $NO_y$ by -2%, -6% and 6%, respectively (Table S1).*"

Line 749: one or more?

Response: We agree with the reviewer's point. We changed the text line 802.

Line 757: see NOy discussion. How is this known?

Response: See our response to the $NO_y$ discussion above.

Line 787: "…higher than…"

Response: Fixed.

Line 936: see $NO_y$ discussion above

Response: See our response to the $NO_y$ discussion above.

Conclusions point 6: Averaging data always results in less scatter. How useful really are instrument comparisons when averaging data spanning orders of magnitude?

Response: Here, we did not average the data but we integrated the signal across each transects (so more analogous to a sum than an average). Comparison of integrated datasets should reduce the scatter that may occur when rapid variations in mixing ratios occur faster than the measurement period and/or with greater spatial heterogeneity than the distance between the sampling locations on a large aircraft.

**References:**

Bradshaw, J., Sandholm, S., and Talbot, R.: An update on reactive odd-nitrogen measurements made during recent NASA Global Tropospheric Experiment programs, J. Geophys. Res. Atmospheres, 103, 19129–19148, https://doi.org/10.1029/98JD00621, 1998.

Cazorla, M., Wolfe, G. M., Bailey, S. A., Swanson, A. K., Arkinson, H. L., and Hanisco, T. F.: A new airborne laser-induced fluorescence instrument for in situ detection of formaldehyde throughout the troposphere and lower stratosphere, Atmospheric Meas. Tech., 8, 541–552, https://doi.org/10.5194/amt-8-541-2015, 2015.

Robinson, M. A., Neuman, J. A., Huey, L. G., Roberts, J. M., Brown, S. S., and Veres, P. R.: Temperature dependent sensitivity of iodide chemical ionization mass spectrometers, EGUsphere, 1–17, https://doi.org/10.5194/amt-2022-295, 2022.

Warneke, C., Schwarz, J. P., Dibb, J., Kalashnikova, O., Frost, G., Seidel, F., Al-Saadi, J., Brown, S. S., Washenfelder, R., Brewer, A., Moore, R. H., Anderson, B. E., Yacovitch, T., Herndon, S., Liu, S., Jaffe, D., Johnston, N., Selimovic, V., Yokelson, B., Giles, D., Holben, B., Goloub, P., Popovici, I., Trainer, M., Pierce, B., Fahey, D., Roberts, J., Soja, A., Peterson, D., Saide, P. E., Holmes, C., Wang, S., Coggon, M. M., Decker, Z. C. J., Ye, X., Stockwell, C., Xu, L., Gkatzelis, G., Lefer, B., Crawford, J. Fire Influence on Regional to Global Environments and Air Quality (FIREX-AQ). To be submitted. 2022.

---

## Author Comment (AC2)

We thank the reviewers for their time and constructive comments that have improved our manuscript. Below we include specific responses to the reviewer's comments. The reviewer's comments are in black. Authors' responses are in blue, quotes from the manuscript are in *italic*, and changes to the text are shown in red.
* * *
Bourgeois et al. report aircraft measurements made on board the NASA DC-8 during the FIREX-AQ campaign in 2019. In this paper, the authors compare duplicate measurements of NO mixing ratios by chemiluminescence and by laser-induced fluorescence (LIF), of $NO_2$ by photolysis coupled to CL (P-CL), cavity-enhanced absorption spectroscopy (CES), and LIF, HONO by chemical ionization mass spectrometry (CIMS) and CES, and of CO by tunable diode laser absorption spectroscopy (TLDAS) and integrated cavity output spectroscopy (ICOS). The authors also attempt to close the $NO_y$ budget by comparing $NO_y$ measured by CL with a sum of individually measured components, $\Sigma NO_y$, calculated by adding NO, $NO_2$, HONO, $HNO_3$ (measured by another CIMS), $pNO_3$ (measured using an aerosol mass spectrometer, AMS) and acyl peroxynitrates (APNs) that were quantified by a third CIMS.

This is a well written manuscript though perhaps a bit too long. There is a lot of interesting results, for example, a great validation of the new LIF instrument and excellent agreements for NO and $NO_2$, but there were also a few questionable items (see below) that the authors will hopefully be able to address in the finalization of this manuscript.

General/Major comments

(1) Tables are, strangely, absent from this paper. Having tables would have helped consolidate this rather long manuscript. Specifically:

Please add a table of measurements/instruments.

Please also add a table of the flight schedule(s), indicating time of day and whether there were nighttime flights analyzed here.

Please add a table which summarizing statistics on the mixing ratios observed (e.g., median, average, percentiles, max and min etc.).

Please consolidate the various correlation slopes/intercepts in one or more tables as well.

Response: We thank the reviewer for the suggestions. We have added a table that lists measurements, instruments and uncertainties (now Table 1). After careful consideration, we decided not to add any additional table to the manuscript for the following reasons:

- a table of flights schedules, including date and time will be provided in the FIREX-AQ overview paper (Warneke et al. 2022), that has been referenced in the manuscript in response to Reviewer 1.

- a table with mixing ratios statistics would be difficult to provide since those statistics depend on the environmental conditions (e.g., smoke vs background, aged smoke vs fresh smoke, wildfires vs eastern fires) and also on the instruments. Most important for the manuscript is the range of mixing ratios being sampled, which are well shown in the correlation plots such as Figures 2, 7, 8, 9 and 14.

- a table with the various correlation slopes/intercepts would not bring new information to the manuscript but add to its (already considerable) length.

| SPECIES | INSTRUMENT | UNCERTAINTY |
|---|---|---|
| NO | CL | $\pm$ (4 % + 6 pptv) |
| | LIF | $\pm$ (8 % + 1 pptv) |
| $NO_2$ | CL | $\pm$ (7 % + 20 pptv) |
| | CES | $\pm$ (5% + 0.26 ppbv) |
| | LIF | $\pm$ (10% + 100 pptv) |
| HONO | CIMS | $\pm$ (15% + 3 pptv) |
| | CES | $\pm$ (9% + 0.6 ppbv) |
| $NO_y$ | CL | $\pm$ (12 % + 15 pptv) |
| | Sum | ~ 25% |
| CO | TD-LAS | 2–7% |
| | ICOS | $\pm$ (2.0 ppb + 2%) |

**Table 1** List of measured species and instruments, including the corresponding uncertainties, during FIREX-AQ

(2) Please clarify if the comparisons made here were "blind" or if kibitzing was allowed/possible before individual PIs reported their data.

Response: The design of FIREX-AQ was to conduct a scientific campaign where a double blind intercomparison between measurement techniques was not the main objective. Several individual measurements of important species were present on the research aircraft during the campaign, and we seized the opportunity to conduct this intercomparison. All PIs had access to other instruments measurements throughout the campaign, so we could not qualify the comparisons as "blind".

We added a sentence to the text lines 568-569 to clarify this point:

"*Comparisons in this manuscript are not blind as all PIs had access to other instruments measurements throughout the campaign.*"

(3) Some instrument descriptions are very thorough (and thank you for that!) yet important details are missing for others. For example, APN data presented, but it is unclear which individual compounds were actually quantified (PAN, PPN, MPAN, APAN etc.) and included in the sum. There was also no statement as to how good or uncertain these data are. HCN and $NH_3$ concentrations were quantified (Figure S14) but their measurement is not described at all.

Response: Thank you for raising this point. We added a description of individual APNs compounds that were quantified as well as the associated measurement uncertainties lines 425-428.

"*APNs species measured during FIREX-AQ include PAN, acryloyl peroxynitrate (APAN), propionyl peroxynitrate (PPN), and peroxybenzoyl nitrate (PBN) with an uncertainty of 20%, 30%, 30% and 30%, respectively.*"

We also included a brief description and appropriate references of HCN and NH$_3$ measurements in the caption of Figure S14 (now Figure S16).

"*HCN was measured by CIMS* (Crounse et al., 2009, 2006). *NH$_3$ was measured by PTR-MS* (Norman et al., 2007)."

In addition, we added the description of the HCN measurement in the Caltech CIMS instrument section lines 376-394. We did not include a description of the NH$_3$ measurement in the main text for the sake of brevity and also because NH$_3$ measurements are presented in Figure S14 (now Figure S16) only.

"*Observations of HNO$_3$, HCN, and hydroxyl nitrates produced from the oxidation of ethane, propene, butane, and isoprene were made by the California Institute of Technology Chemical Ionization Mass Spectrometer (CIT-CIMS) compact time-of-flight (cToF, TofWerk/Caltech) sensor using CF$_3$O$^-$ ion chemistry* (Crounse et al., 2006). *In short, a large flow of ambient air (about 40 m$^3$ s$^{-1}$) was rapidly brought into the aircraft through a Teflon coated glass inlet (warmed slightly above ambient temperature), where it was subsampled, diluted with dry N$_2$, reacted with CF$_3$O$^-$, and underwent subsequent product ion analysis by time-of-flight mass spectrometry. The HF•NO$_3^-$ (m/z 82) product ion is used to quantify HNO$_3$. The HCN and hydroxy nitrates are detected as cluster ions. Laboratory-generated, T-dependent and water-dependent calibration curves were performed to produce ambient mixing ratios from raw signals for HNO$_3$ and hydroxy nitrates The HCN sensitivity is tracked in situ based on the continuous addition of isotopically labeled H$^{13}$C$^{15}$N into the instrument from a custom-made gravimetrically based compressed gas cylinder. In-flight instrumental zeros were performed every ~15 minutes using dry N$_2$ and ambient air passed through NaHCO$_3$-coated nylon wool. Continuous data, with the exception of zero and calibration periods, are reported with 1Hz frequency. The uncertainties for HNO$_3$, HCN, and hydroxy nitrates are ± (30% + 50 pptv), ± (25% + 70 pptv), and ± (25% + 3 pptv), respectively.*"

(4) Measurements of HNO$_3$, APNs, ClNO$_2$, N$_2$O$_5$, pNO$_3$, C1-C5 alkyl nitrates were made but sample time series of those data are not shown, which is an odd omission considering that some of these compounds contribute the most to NO$_y$ (judging from Figure 10).

Response: We added two supplemental figures to the SI that show timeseries of HNO$_3$, APNs, ClNO$_2$, N$_2$O$_5$ and pNO$_3$ (Figures S1 and S2).

[Figure]

**Figure S1** 1 s measurements of a) Isoprene hydroxy nitrate (ISOPN) and $C_1$–$C_5$ alkyls nitrates (ANs), b) particulate nitrate ($pNO_3$) and $HNO_3$, c) $N_2O_5$ and $ClNO_2$ and d) APNs during two crosswind plume transects of smoke from the Williams Flat fire on 07/08/2019. The plume transects were chosen due to the significant enhancement of all species at that time.

[Figure]

**Figure S2** 1 s measurements of a) Isoprene hydroxy nitrate (ISOPN) and $C_1$–$C_5$ alkyls nitrates (ANs), b) particulate nitrate ($pNO_3$) and $HNO_3$, c) $N_2O_5$ and $ClNO_2$ and d) APNs during crosswind plume transects of smoke from crop burning in southeastern US on 30/08/2019.

(5) The definition and choices/explanations as to what species to include in $\Sigma NO_y$ in this manuscript (abstract line 14; equation 2, line 339) would benefit from some polishing.

(a) Definitions.

Please add (to the introduction - see comment on lines 95-98) a comprehensive definition of what species contribute to $NO_y$ (e.g., equation (1) of Fahey et al., J. Geophys. Res., 91, 9781-9793, 10.1029/JD091iD09p09781, 1986), if only to provide a contrast to equation (2) of this manuscript.

Many components of $NO_y$ are omitted from equation (2). Please note more prominently the (many) omissions from $\Sigma NO_y$ in the abstract, such as higher molecular weight alkyl nitrates ("total alkyl nitrates", line 846), coarse nitrate, peroxynitrates ($HO_2NO_2$, $RO_2NO_2$), and the nocturnal nitrogen oxides $NO_3$, $N_2O_5$ and $ClNO_2$.

Since the expression given here for $\Sigma NO_y$ is a simplification, the right-hand side of equation (2) only approximates $\Sigma NO_y$ and an equal sign should not be used (use $\approx$ instead).

Further, since the expression for $\Sigma NO_y$ omits nocturnal nitrogen oxides, the definition of $\Sigma NO_y$ as in equation (2) should perhaps be referred to as the sum of daytime nitrogen oxides, and the time of day of the measurements should be added to the title.

Response: We thank the reviewer for the great suggestions. We added the definition from Fahey et al. 1986 to the introduction lines 99-109.

*"Fahey et al. (1986) define $\Sigma NO_y$ as the sum of important nitrogen species as illustrated by Eq. 1:*

*$\Sigma NO_y = NO + NO_2 + $ nitric acid $(HNO_3) + HONO + $ peroxynitric acid $(HO_2NO_2) + $ nitrate $(NO_3) + $ dinitrogen pentoxide $(2*N_2O_5) + $ peroxyacetyl nitrate $(PAN) + $ particulate nitrate $(pNO_3) + ...$*         *(Eq. 1)*

*Other nitrogen compounds that can contribute to $\Sigma NO_y$ include alkyl nitrates* (Day et al., 2003)*, acyl peroxynitrates (APNs; Juncosa Calahorrano et al., 2021), non-acyl peroxynitrates $(RO_2NO_2$ ; Murphy et al., 2004), nitryl chloride $(ClNO_2$ ; Kenagy et al., 2018), nitro compounds and nitroaromatics (Decker et al., 2021)."*

As suggested, we clarified the definition of $\Sigma NO_y$ in the abstract lines 15-19.

*"Other $NO_y$ species were not included in $\Sigma NO_y$ as they either contributed minimally to it (e.g., $C_1$-$C_5$ alkyl nitrates, nitryl chloride $(ClNO_2)$, dinitrogen pentoxide $(N_2O_5))$ or were not measured during FIREX-AQ (e.g., higher oxidized alkyl nitrates, nitrate $(NO_3)$, non-acyl peroxynitrates, coarse mode aerosol nitrate)."*

We also used the recommended formalism in Equation 2 line 361.

*"$\Sigma NO_y \approx NO_x + HONO + HNO_3 + pNO_3 + APNs$   (Eq. 2)"*

(b) Organization.

It is clear from the outset that several components of $NO_y$ were measured by multiple instruments, yet the reader is kept in the dark for far too long what the authors included in this sum and what they mean by $\Sigma NO_y$ (e.g., line 14 and 339). If I counted correctly, there are (at least) 36 different ways $\Sigma NO_y$ could have possibly been calculated for this data set (NO from either one of two instruments or average NO which gives 3 possibilities, $NO_2$ from one of three instruments or average $NO_2$ to give 4 possibilities, HONO from one of two instruments or average HONO to give 3 possibilities, $3\times4\times3 = 36$ possible combinations). The reader is only told on line 732 which measurements were actually used.

Response: We thank the reviewer for raising this issue. We moved the description of instruments that were used to calculate $\Sigma NO_y$ to section 2.2.6 lines 368-374.

(c) Closure.

Having so many choices (data from several instruments to choose from, and which compounds to include in $\Sigma NO_y$) is great, but ultimately undermines the conclusion that $NO_y$ budget closure was achieved (lines 22/23).

Even though I know this wasn't the case, the manuscript somehow gave me the vibe that data were cherry-picked and the authors stopped adding compounds to $\Sigma NO_y$ once the slope relative to $NO_{y,CL}$ reached unity. Can you be more convincing - for example, why not add all components that were quantified - surely, there would have been times when all instruments

were operational? Please add such a plot (and use the larger $NO_x$ and HONO data from the LIF & CES instruments).

Response: As explained in the manuscript lines XXX, other nitrogen oxides (alkene hydroxy nitrates, nitromethane ($CH_3NO_2$), $N_2O_5$, $ClNO_2$, and $C_1$–$C_5$ alkyl nitrates) were also measured during FIREX-AQ but were not included in this equation as they contributed on average less than 7% to the $NO_y$ budget (see section 3.4). Further, including these measurements would have decreased data availability for comparison with the total $NO_y$ measurement by more than 60%.

We added a plot (Figure S14) showing the $NO_y$ comparison when all measurements were available (except for $C_1$-$C_5$ alkyl nitrates as those are discrete measurements). We also added a discussion to the text lines 786-789.

"*Including minor $NO_y$ species (= $ClNO_2$, $N_2O_5$, $CH_3NO_2$, and alkene hydroxy nitrates) in the $\Sigma NO_y$ had little effect on the correlation between $\Sigma NO_y$ and CL $NO_y$ and resulted in a slope of $1.02 \pm 0.25$ ($R^2 =0.94$) and an intercept of $-0.68 \pm 0.01$ ppbv (Figure S14).*"

[Figure]

Figure S14: Comparison of the sum of individually measured $NO_y$ species (= $NO_x$ + HONO + $HNO_3$ + APNs + $pNO_3$ + alkene hydroxy nitrates + $CH_3NO_2$ + $ClNO_2$ + $N_2O_5$) with the total $NO_y$ measurement by CL. Data from the entire campaign are presented in panels a) and b). Here LIF NO, CES HONO and CES $NO_2$ are used in the sum of $NO_y$.

And please discuss the elephant in the room: The unquantified components of $NO_y$. If closure was indeed achieved, it would imply that those unquantified components were negligible, which in my opinion is doubtful.

It is stated on line 846, that FIREX-AQ did not include a measurement of total alkyl nitrates, but the thought is left hanging. What if the suite of instruments had included such a measurement? Would the $NO_y$ budget have blown up? I'd be surprised if the Cohen group had not quantified $\Sigma AN$ in fire plumes at some point to help constrain this "known unknown" and to guide this discussion.

Response: We thank the reviewer for raising this point. Wolfe et al. (2022) presents total alkyl nitrate measurements from the Rim Fire during the SEAC[4]RS campaign. They find that about 10% of the $NO_y$ budget consists of total alkyl nitrate (ANs), and that ANs are typically one order of magnitude less abundant than peroxynitrates. We added this discussion to the text lines 914-918.

"*A recent analysis of the California Rim Fire during the 2013 NASA Studies of Emissions, Atmospheric Composition, Clouds and Climate Coupling by Regional Surveys (SEAC[4]RS) mission report that total alkyl nitrates measured by TD-LIF accounted for ~10% of the $NO_y$ budget* (Wolfe et al., 2022)."

Also, if submicron $pNO_3$ constituted ~40% or so of $NO_y$ in wildfire plumes (Figure 10a), surely there would have been coarse nitrate as well, which would have consequences on closure. More discussion is needed. There were measurements of coarse mode size distributions (Schoeberl et al., Coarse mode aerosol in biomass burning aerosol layers during FIREX-AQ, TBD, in prep, 2021 - listed on https://csl.noaa.gov/projects/firex-aq/science/pubs.html and Noyes et al., Remote Sensing 12(22), 3223, https://doi.org/10.3390/rs12223823) that may provide some constraints here.

Response: We thank the reviewer for raising this issue. During FIREX-AQ, the large majority of coarse mode aerosols were in the form of ash, not dust, with very little nitrate associated to them (Adachi et al., 2022). Furthermore, a PMF analysis of the flight (on 07/08/2019) with the largest amount of coarse mode particulate calcium and nitrate (as measured by a bulk aerosol sampling system coupled to an ion chromatograph (SAGA instrument) with an approximated $PM_4$ cutoff (Dibb et al., 2002; Brock et al., 2019)) shows that the amount of coarse mode particulate nitrate reported by the SAGA instrument was consistent with the amount of submicron inorganic $pNO_3$ measured by the HR-AMS measurements:

[Figure]

This result indicate that particulate nitrate is almost exclusively in the accumulation mode, with little to none contribution from the coarse mode, which may be explained by the low $HNO_3$ concentrations in the sampled smoke plumes resulting in a slow uptake on coarse aerosols.

During FIREX-AQ, measured coarse mode particulate nitrate was greater than measured submicron $pNO_3$ for ~10% of the data, and only under conditions where $pNO_3$ was a small contributor to the total $NO_y$ (2 $\mu$g sm$^{-3}$ or less total aerosol nitrate). Therefore, the overall contribution of coarse mode particulate nitrate to the $NO_y$ budget is should be minimal.

Additionally, neither the CL instrument nor the HR-AMS instrument measured super-micron aerosol because of inlet cut-off, meaning that coarse mode aerosol nitrate is not accounted for on either side of equation 2 and should not have consequences on $NO_y$ budget closure. We added a sentence to the main text lines 774-779 to reflect this point:

"*Based on comparisons of HR-AMS $pNO_3$ with on-board filters collecting aerosols with a size cut around 4$\mu$m (Brock et al., 2019; Dibb et al., 2002), coarse mode particulate nitrate did not significantly contribute to the total $NO_y$ budget during FIREX-AQ. Additionally, coarse mode particulate nitrate was not measured by either the HR-AMS or the $NO_y$ inlet in the CL instrument and therefore does not contribute to the intercomparison presented here.*"

(6) Carbon monoxide

The sections on CO seem like an afterthought and do not add much to the remainder of the paper. I'd recommend splitting this off into a separate to reduce the size of this already very long paper.

Response: We thank the reviewer for the suggestion. We would like to keep the CO comparison into this paper, as we believe that this comparison fits well in the current manuscript. CO is an essential component to fire science, similar to nitrogen compounds. It provides a reference species that is only affected by dilution on the timescales usually considered when investigating chemistry in smoke plumes (i.e., a couple of hours) and it is extensively used in the calculation of normalized excess mixing ratios as well as important fire parameters such as the modified combustion efficiency. Ensuring that CO was measured accurately during FIREX-AQ is thus crucial to get all following analyses right.

Specific/Minor comments

line 21. a slope of 1.8 - yikes!

Response: We agree with the reviewer that this is a large slope. It is adequately discussed in the manuscript and does not require further justification in the abstract.

line 72. Please add a table summarizing this large suite of airborne instruments.

Response: We have added this table in the figures (Table 1) and a reference to it line 78.

"*During FIREX-AQ, a large suite of airborne instruments, detailed in the following sections, performed independent in situ tropospheric measurements of one or more fire-science relevant reactive nitrogen species and CO aboard the NASA DC-8 aircraft (Table 1).*"

lines 95-98. Please insert an equation here, defining $NO_y$ (similar to equation (1) of Fahey et al., J. Geophys. Res., 91, 9781-9793, 10.1029/JD091iD09p09781, 1986).

Response: Please refer to our response above regarding this point.

line 112. There have been other papers from this campaign (e.g., Decker et al.) that would be worth calling out here.

Response: We thank the reviewer for the suggestion. We added Decker et al. lines 106-109.

"*Other nitrogen compounds that can contribute to $\Sigma NO_y$ include alkyl nitrates (Day et al., 2003), acyl peroxynitrates (APNs; Juncosa Calahorrano et al., 2021), non-acyl peroxynitrates (RO$_2$NO$_2$; Murphy et al., 2004), chlorine nitrite (ClNO$_2$; Kenagy et al., 2018), nitro compounds and nitroaromatics (Decker et al., 2021).*"

lines 159. Pollack et al. describe two converters with LEDs at 365 nm and one converter at 395 nm, but not one at 385 nm. Is this a new system? If so, please provide relative data such as make/power of the LEDs, NO$_2$ photolysis frequency, temperature etc.

Response: We thank the reviewer for the correction. It is a new system – we removed the reference to Pollack et al. and we added the relative data of the LEDS lines 171-173.

"*In the NO$_2$ channel, NO$_2$ is photolyzed to NO with a 40 ± 1 % conversion efficiency using two ultraviolet (UV) LEDs (Hamamatsu, model L11921) at 385 nm in a 45 cm long quartz cell (inner diameter of 1.2 cm) pressure-controlled at 209.8 ± 0.3 Torr.*"

line 160. Pollack et al. - the Journal of Atmospheric Chemistry lists this citation as a 2010 paper (even though it was only accepted in 2011). Please update.

Response: Done.

line 180. "5% HONO interference". The magnitude of this interference will depend on the ratio of HONO to NO$_2$ in ambient air. Please clarify what is meant by 5% (stated on lines 615-617: 5% of the HONO sampled converts to NO).

Response: We mean that at a wavelength of 385nm, about 5% of the HONO signal will be converted into NO$_2$ and cause an interference in the NO$_2$ measurement. We corrected the text lines 193-194 to clarify this aspect of the instrument description.

"*Finally, NO$_2$ data were further corrected for a HONO interference (5% of the HONO mixing ratios) due to HONO photolysis at 385 nm quantified from theoretical calculation and confirmed in the laboratory using a HONO source described in Lao et al. (2020).*"

line 209. please provide an uncertainty estimate for the NO-LIF instrument similar to lines 183, 220 and 280.

Response: We thank the reviewer for the suggestion. We added the estimate lines 225-226 and also in Table 1.

"*The NO measurement uncertainty is estimated to be ± (8% + 1 pptv).*"

line 247. please state how the zero air was generated (cylinder or scrubbed air).

Response: We used zero air from a cylinder, and we added the correction to the text lines 266 and 279.

line 259. Please state how often the Teflon filters were changed.

Response: The filters were changed prior to each flight. We clarified this point in the manuscript line 278.

line 271. a 0th order polynomial - interesting way to say "offset".

Response: The polynomial order can be something other than 0, but was zero in this case. Thus this term is more accurate than the term "offset".

lines 270-276. Please comment on errors introduced from using reference absorption cross-sections are measured at near 1 atm pressure and near room temperature to fit absorption spectra collected at reduced pressure and ambient (I am guessing) temperature.

Response: The absorption spectra are not pressure dependent, and measurements took place at ambient temperature inside the cabin. Therefore, we don't expect the cross sections to change under our experimental conditions, and we don't anticipate any additional error here.

line 281. What is the effective optical path of this instrument?

Response: During FIREX-AQ, the effective path length was about 5.3 km. We do not provide this information in the manuscript as instrument precision is already described.

line 307. What is the linear dynamic range of this instrument?

Response: The dynamic range for the instrument is species dependent. For compounds that are measured with high sensitivity, the dynamic range is the smallest, since the product ions may deplete the reagent ions. For high sensitivity compounds, the response remains linear up to tens of ppbv. For HONO, which is measured with relatively lower sensitivity, the response remains linear up to 100s of ppbv.

line 310. "normalized by the iodide signals" - $I^-$ or $I^- \cdot H_2O$ or both? The Pratt group has recently used the water cluster to normalize.

Response: We normalize by IH2O, we added the precision in the text.

line 313-314. "Calibrations with $Cl_2$ and $HNO_3$ permeation sources ... to diagnose the stability of instrument sensitivity" - please comment on how stable that response turned out to be (perhaps further down in the results section).

Response: The standard deviation of inflight calibrations is typically 10%. We added this information lines 334-335.

line 321. background typically equivalent to 40 ppt - what was the range of backgrounds observed? Does the background increase after sampling high concentrations of HONO?

Response: The temperature dependence to sensitivity results in background variations roughly from 10 to 160 ppt. Importantly, the backgrounds did not increase following sampling of

concentrated fire plumes. There is no evidence that HONO, nor any other compound, sticks to the instrument surfaces and later desorbs.

line 339. Data from which instruments were used to account for the species in equation (2)?

Response: We moved the description of instruments used in equation 2 to section 2.2.6 lines 368-370.

line 372-373. Can you speculate how much coarse nitrate there might be in a biomass burning plume?

Response: Please refer to our response above to comment #5.

line 393. please provide an uncertainty estimate for the CIMS measuring APNs instrument similar to lines 183, 220 and 280 (see also comment for line 209).

Response: We added the uncertainty estimate lines 425-428.

line 404-415. Are the $N_2O_5$ data presented anywhere? If these data are from the same instrument that underestimated HONO by a factor of 1.8, how confident can one be in the $N_2O_5$ data and stated $\pm(15\% + 2$ pptv) accuracy?

Response: We now present $N_2O_5$ and $ClNO_2$ data in Figures S1 and S2. As described in Robinson et al. (2022), the temperature dependence does not affect $N_2O_5$.

line 431. "at approximately 4.6 μm" Since these types of instruments monitor a specific absorption line and derive mole fractions based on that particular line's line strength, please be more specific here. In general, more detail (or a more appropriate citation) is needed in this section since the Baer et al. (2002) reference does not describe an instrument quantifying CO via its absorption in the mid-IR.

Response: The ICOS instrument measures CO at 2190.0cm$^{-1}$, or 4.566μm. We modified the text accordingly and we added an additional reference (Arévalo-Martínez et al., 2013) to the manuscript lines 465-467:

"*CO was measured using a modified commercial off-axis ICOS instrument (Los Gatos Research (LGR) N₂O/CO-30-EP; Arévalo-Martínez et al., 2013; Baer et al., 2002) at 4.566 μm.*"

line 442 and 456-457 "dry air mole fraction". Is this correction made purely because the water vapour variability is sufficiently large to cause deviations to mole fractions, or are there other effects in play, too, such as spectral broadening or overlap with water lines in the IR? Please add an explanation and justification for this correction to the text.

In practice, how much of a correction was made, and perhaps most importantly, why were only the ICOS data corrected and not also the TDLAS instrument described in 2.2.8 which used an absorption line ~4.7 μm and whose data would have equally been affected by the presence of water vapor?

Response: Both instruments report dry air mole fraction. The correction is made for displacement purposes, since the water vapor mixing ratio is often on par with the uncertainty of the measurement. We modified the text line 477.

"(...) *(both ICOS-CO and TDLAS-CO* mixing ratios are reported as dry air mole fractions)."

line 451. "precision" - is that for 1-second data?

Response: Yes, this is the 1-Hz precision. We clarified the text line 486.

"*The 1-Hz precision of the measurement in flight is estimated to be 0.4 ppb.*"

line 533. Please cite a paper for orthogonal distance regression or describe the algorithm.

Response: We added a reference for Orthogonal Distance Regression line 572.

"*We first calculated the slope of the linear least-squares (LLS) orthogonal distance regression (ODR; Boggs et al., 1987) to characterize the percent difference between measurements of a pair of instruments weighted by the inverse of the instrument precision.*"

line 556. Figure 2a shows a slope of 0.98±0.00 whereas the text has 0.98±0.08. The meaning of the error is defined for the text (±combined instrument uncertainties) but not for the Figures since the values there are different. Please clarify.

Also, please state how combined uncertainties were calculated.

Response: We updated the figures to reflect an error that corresponds to that defined in the text. Combined uncertainties were calculated be adding in quadrature individual instrument uncertainties. We added a sentence to clarify this point lines 589-591.

"*In the following sections, combined instrument uncertainties were calculated by adding in quadrature individual instrument uncertainties.*"

lines 554 - 577. Impressive performance by a new instrument! Well done!

Response: Thank you.

line 609. "ranging from 0.88±0.12 to 0.90±0.11". This large difference is interesting. Wouldn't that suggest that the CL $NO_y$ data may also be 10% - 12% too low, since it would have been calibrated using $NO_x$ calibration standards?

Response: It is unlikely that the difference between CL $NO_2$ and other $NO_2$ measurements was due to a calibration issue. If so, the CL NO measurement, which was calibrated using the same standard as for the CL $NO_2$ measurement, would also have been 10-12% higher than the NO LIF measurement (which was calibrated using an independent standard). This was not the case during FIREX-AQ (see section 3.1). Therefore, there is no reason to suspect a calibration error in the CL $NO_y$ measurement.

We added this discussion to the main text lines 654-659.

"*However, it is unlikely that the difference between CL $NO_2$ and other $NO_2$ measurements was due to a calibration issue. If so, the CL NO measurement, which was calibrated using the same standard as for the CL $NO_2$ measurement, would also have been 10-12% higher than the NO LIF measurement (which was calibrated using an independent standard). This was not the case during FIREX-AQ (see section 3.1).*"

line 609. "comparable" is probably not the best word in this context - suggestion: "on the upper end of the combined uncertainties" or similar.

Response: We modified the text accordingly.

line 618. how much HONO was there relative to $NO_2$?

Response: HONO to $NO_2$ ratio was typically between 0.2–0.4 during FIREX-AQ. 5% of that ratio means that at most 2% of the $NO_2$ signal was due to HONO interference. We added this precision in the text lines 660-664.

"*However, this interference was determined to be low (less than 5% of HONO concentration; typical HONO to $NO_2$ ratios ranged between 0.2-0.4 during FIREX-AQ) following laboratory tests using a HONO calibration source* (Lao et al., 2020)*, and the $NO_2$ measurement by CL was corrected for it*"

lines 666-697. Sounds like the CIMS would benefit from an internal standard to track its HONO sensitivity, e.g., continuous addition of a calibrated amount of $^{15}N^{18}O_2H$ to the inlet.

If I understood this correctly, one HONO instrument sampled through a filter, the other did not. Please comment on what role, if any, the filter on the CES may have played? There are indications that $NO_2$ can convert on surface to HONO. Has the CES inlet transmission of $NO_2$ been tested using an "aged" filter?

Response: Yes, the CES had a filter. But CES-HONO tended to be higher than CIMS-HONO during FIREX-AQ. If the filter were causing transmission loss, then CES-HONO would have been lower. We did not test an "aged" filter, but there wasn't any trend in $\Delta NO_{2CES-CL}$ or $\Delta HONO_{CES-CIMS}$ with flight time which indicates no significant loss on the filters. Additionally, filters were changed prior to each flight (see response above).

line 720. "$NO_y$". Usually, $NO_x$ constitutes the largest fraction of $NO_y$. Since there was good agreement between $NO_x$ measurements, good agreement can also be expected for $NO_y$. Consider a section on $NO_z = NO_y − NO_x$.

Response: This is true in most urban settings. However, in smoke plumes $pNO_3$ and APNs rapidly become the most prevalent $NO_y$ components. For this reason, adding a section on $NO_z$ would actually be redundant with the current section on $NO_y$.

line 723. Section 2.2.8 should be section 2.2.6.

Response: Fixed.

line 817. How were HCN and NH$_3$ quantified?

Response: We added a brief instrument description and appropriate reference in the caption of Figure S16 (see also our comment above).

line 817. "Here, we find no evidence for a potential interference of HCN or NH$_3$" - thats' good news! Is there an explanation as to why this instrument outperforms others in this regard?

Response: NH$_3$ and HCN interferences in CL instruments have been demonstrated in laboratory settings and in dry air conditions (Fahey et al., 1985). However, the same study showed that those interferences accounted for less than 1% of measured NO$_y$ for air of 20% relative humidity. During FIREX-AQ, ambient air relative humidity was typically higher than 20% (average value of 37%), so negligible interference from those compounds were expected.

line 846. "However, FIREX-AQ did not include a measurement of total alkyl nitrates." And if it had, would the result have been $\Sigma NO_y >> NO_{y,CL}$? I wonder ...

Response: Please refer to our response above to comment #5

line 953. My browser displayed: "Hmm. We're having trouble finding that site." Please verify the link to the archive.

Response: Fixed.

Figures 2a, 9a, and 12a. Are all data included in these panels, or a selection? Please clarify in the caption(s).

Response: All data are included in panels 2a and 9a, and we clarified this in the captions. However, as stated in Figure 12a caption, the data shown is from one individual fire smoke (Williams Flat fire on 08/07).

Figure 3. Please clarify in the caption at what time of day these plumes were observed (>20 ppbv of daytime HONO would seem like a lot during daytime).

Response: The local time is given in the x-axis of Figure 3. The smoke plume form the Williams Flat fire sampled on 08/07 (presented in Figure 3) was wide and thick, creating "nocturnal" conditions at the heart of the plume. This explained the elevated mixing rations of HONO sampled during daytime.

Figure 8. Since the CES data are likely more accurate, consider switching the axes (plotting CIMS vs CES data). Were photolysis frequencies quantified? Are these daytime HONO levels? If there was truly this much HONO in the daytime, more justification as to the suggested absence of other photolabile compounds (N$_2$O$_5$/ClNO$_2$) is needed.

Response: We thank the reviewer for raising this point. See previous comment regarding the time of the day. Decker et al., 2021 recently showed that in smoke plumes NO$_3$ reactivity is largely dominated by VOCs leading to the production of nitro-aromatics or HNO$_3$. As a result, there was little to no formation of N$_2$O$_5$ and ClNO$_2$ in smoke during FIREX-AQ (see Figures S& and S2).

Figure 10. Please state what percentiles are used of the box-and-whisker plots.

Response: Done.

Supplement

The figures here are labeled SA, SB, SC, ... and S1, S2, S3, but could have just been numbered consecutively to avoid unnecessary confusion.

Response: We thank the reviewer for the suggestion. We'd like to keep the numbering as it is as Figures SA-SD belong with the modeling work on $pNO_3$ transmission in the $NO_y$ inlet rather than with the main text.

Figure S12. I am surprised not to see a larger difference in the slopes of Figures S12a and 9c, considering $NO_x$ (~30% of $NO_y$ in background air judging from Figure 10) would have been increased by 10%-12% and HONO (which was abundant at times also - Figure 8) by 80%, yet the slopes are virtually identical (1.00±0.01 and 1.01±0.00). Since a distinction was made in Figure 10 between background air and "in smoke", please also make that distinction in Figures 9 and S12.

Response: Using CES NO2 actually decreases the slope by 6%, while using CES HONO increases the slope by 6%. Using NO LIF decreases the slope by 2%. As a consequence, the slope shown in Figure S12 (now Figure S14), where CES HONO, CES NO2 and NO LIF were used in the sum of $NO_y$ did not change compared to Figure 9. In smoke, using LIF NO, CES $NO_2$ and CES HONO as primary measurements changed the correlation slope between $\Sigma NO_y$ and measured $NO_y$ by -1%, -8% and 9%, respectively. We added Table S1 where we provide the various slopes calculated depending on the instrument used. We also added a clarification in the main text lines 370-374:

"*Using LIF NO, CES $NO_2$ and CES HONO as primary measurements changed the correlation slope between $\Sigma NO_y$ and measured $NO_y$ by -2%, -6% and 6%, respectively (Table S1). In smoke, using LIF NO, CES $NO_2$ and CES HONO as primary measurements changed the correlation slope between $\Sigma NO_y$ and measured $NO_y$ by 1%, -8% and 9%, respectively (Table S1).*"

**References:**

Adachi, K., Dibb, J. E., Scheuer, E., Katich, J. M., Schwarz, J. P., Perring, A. E., Mediavilla, B., Guo, H., Campuzano-Jost, P., Jimenez, J. L., Crawford, J., Soja, A. J., Oshima, N., Kajino, M., Kinase, T., Kleinman, L., Sedlacek III, A. J., Yokelson, R. J., and Buseck, P. R.: Fine Ash-Bearing Particles as a Major Aerosol Component in Biomass Burning Smoke, J. Geophys. Res. Atmospheres, 127, e2021JD035657, https://doi.org/10.1029/2021JD035657, 2022.

Arévalo-Martínez, D. L., Beyer, M., Krumbholz, M., Piller, I., Kock, A., Steinhoff, T., Körtzinger, A., and Bange, H. W.: A new method for continuous measurements of oceanic and atmospheric $N_2O$, CO and $CO_2$: performance of off-axis integrated cavity output

spectroscopy (OA-ICOS) coupled to non-dispersive infrared detection (NDIR), Ocean Sci., 9, 1071–1087, https://doi.org/10.5194/os-9-1071-2013, 2013.

Baer, D. S., Paul, J. B., Gupta, M., and O'Keefe, A.: Sensitive absorption measurements in the near-infrared region using off-axis integrated-cavity-output spectroscopy, Appl. Phys. B, 75, 261–265, https://doi.org/10.1007/s00340-002-0971-z, 2002.

Boggs, P. T., Byrd, R. H., and Schnabel, R. B.: A Stable and Efficient Algorithm for Nonlinear Orthogonal Distance Regression, SIAM J. Sci. Stat. Comput., 8, 1052–1078, https://doi.org/10.1137/0908085, 1987.

Brock, C. A., Williamson, C., Kupc, A., Froyd, K. D., Erdesz, F., Wagner, N., Richardson, M., Schwarz, J. P., Gao, R.-S., Katich, J. M., Campuzano-Jost, P., Nault, B. A., Schroder, J. C., Jimenez, J. L., Weinzierl, B., Dollner, M., Bui, T., and Murphy, D. M.: Aerosol size distributions during the Atmospheric Tomography Mission (ATom): methods, uncertainties, and data products, Atmospheric Meas. Tech., 12, 3081–3099, https://doi.org/10.5194/amt-12-3081-2019, 2019.

Crounse, J. D., McKinney, K. A., Kwan, A. J., and Wennberg, P. O.: Measurement of Gas-Phase Hydroperoxides by Chemical Ionization Mass Spectrometry, Anal. Chem., 78, 6726–6732, https://doi.org/10.1021/ac0604235, 2006.

Crounse, J. D., DeCarlo, P. F., Blake, D. R., Emmons, L. K., Campos, T. L., Apel, E. C., Clarke, A. D., Weinheimer, A. J., McCabe, D. C., Yokelson, R. J., Jimenez, J. L., and Wennberg, P. O.: Biomass burning and urban air pollution over the Central Mexican Plateau, Atmos Chem Phys, 16, 2009.

Day, D. A., Dillon, M. B., Wooldridge, P. J., Thornton, J. A., Rosen, R. S., Wood, E. C., and Cohen, R. C.: On alkyl nitrates, O3, and the "missing NOy," J. Geophys. Res. Atmospheres, 108, https://doi.org/10.1029/2003JD003685, 2003.

Decker, Z. C. J., Robinson, M. A., Barsanti, K. C., Bourgeois, I., Coggon, M. M., DiGangi, J. P., Diskin, G. S., Flocke, F. M., Franchin, A., Fredrickson, C. D., Gkatzelis, G. I., Hall, S. R., Halliday, H., Holmes, C. D., Huey, L. G., Lee, Y. R., Lindaas, J., Middlebrook, A. M., Montzka, D. D., Moore, R., Neuman, J. A., Nowak, J. B., Palm, B. B., Peischl, J., Piel, F., Rickly, P. S., Rollins, A. W., Ryerson, T. B., Schwantes, R. H., Sekimoto, K., Thornhill, L., Thornton, J. A., Tyndall, G. S., Ullmann, K., Van Rooy, P., Veres, P. R., Warneke, C., Washenfelder, R. A., Weinheimer, A. J., Wiggins, E., Winstead, E., Wisthaler, A., Womack, C., and Brown, S. S.: Nighttime and daytime dark oxidation chemistry in wildfire plumes: an observation and model analysis of FIREX-AQ aircraft data, Atmospheric Chem. Phys., 21, 16293–16317, https://doi.org/10.5194/acp-21-16293-2021, 2021.

Dibb, J. E., Talbot, R. W., Seid, G., Jordan, C., Scheuer, E., Atlas, E., Blake, N. J., and Blake, D. R.: Airborne sampling of aerosol particles: Comparison between surface sampling at Christmas Island and P-3 sampling during PEM-Tropics B, J. Geophys. Res. Atmospheres, 107, PEM 2-1-PEM 2-17, https://doi.org/10.1029/2001JD000408, 2002.

Fahey, D. W., Eubank, C. S., Hübler, G., and Fehsenfeld, F. C.: Evaluation of a catalytic reduction technique for the measurement of total reactive odd-nitrogen NO y in the atmosphere, J. Atmospheric Chem., 3, 435–468, 1985.

Fahey, D. W., Hübler, G., Parrish, D. D., Williams, E. J., Norton, R. B., Ridley, B. A., Singh, H. B., Liu, S. C., and Fehsenfeld, F. C.: Reactive nitrogen species in the troposphere: Measurements of NO, NO2, HNO3, particulate nitrate, peroxyacetyl nitrate (PAN), O3, and total reactive odd nitrogen (NO y ) at Niwot Ridge, Colorado, J. Geophys. Res. Atmospheres, 91, 9781–9793, https://doi.org/10.1029/JD091iD09p09781, 1986.

Juncosa Calahorrano, J. F., Lindaas, J., O'Dell, K., Palm, B. B., Peng, Q., Flocke, F., Pollack, I. B., Garofalo, L. A., Farmer, D. K., Pierce, J. R., Collett, J. L., Weinheimer, A., Campos, T., Hornbrook, R. S., Hall, S. R., Ullmann, K., Pothier, M. A., Apel, E. C., Permar, W., Hu, L., Hills, A. J., Montzka, D., Tyndall, G., Thornton, J. A., and Fischer, E. V.: Daytime Oxidized Reactive Nitrogen Partitioning in Western U.S. Wildfire Smoke Plumes, J. Geophys. Res. Atmospheres, 126, e2020JD033484, https://doi.org/10.1029/2020JD033484, 2021.

Kenagy, H. S., Sparks, T. L., Ebben, C. J., Wooldrige, P. J., Lopez-Hilfiker, F. D., Lee, B. H., Thornton, J. A., McDuffie, E. E., Fibiger, D. L., Brown, S. S., Montzka, D. D., Weinheimer, A. J., Schroder, J. C., Campuzano-Jost, P., Day, D. A., Jimenez, J. L., Dibb, J. E., Campos, T., Shah, V., Jaeglé, L., and Cohen, R. C.: NOx Lifetime and NOy Partitioning During WINTER, J. Geophys. Res. Atmospheres, 123, 9813–9827, https://doi.org/10.1029/2018JD028736, 2018.

Lao, M., Crilley, L. R., Salehpoor, L., Furlani, T. C., Bourgeois, I., Neuman, J. A., Rollins, A. W., Veres, P. R., Washenfelder, R. A., Womack, C. C., Young, C. J., and VandenBoer, T. C.: A portable, robust, stable, and tunable calibration source for gas-phase nitrous acid (HONO), Atmospheric Meas. Tech., 13, 5873–5890, https://doi.org/10.5194/amt-13-5873-2020, 2020.

Murphy, J. G., Thornton, J. A., Wooldridge, P. J., Day, D. A., Rosen, R. S., Cantrell, C., Shetter, R. E., Lefer, B., and Cohen, R. C.: Measurements of the sum of $HO_2NO_2$ and $CH_3O_2NO_2$ in the remote troposphere, Atmospheric Chem. Phys., 4, 377–384, https://doi.org/10.5194/acp-4-377-2004, 2004.

Norman, M., Hansel, A., and Wisthaler, A.: O2+ as reagent ion in the PTR-MS instrument: Detection of gas-phase ammonia, Int. J. Mass Spectrom., 265, 382–387, https://doi.org/10.1016/j.ijms.2007.06.010, 2007.

Robinson, M. A., Neuman, J. A., Huey, L. G., Roberts, J. M., Brown, S. S., and Veres, P. R.: Temperature dependent sensitivity of iodide chemical ionization mass spectrometers, EGUsphere, 1–17, https://doi.org/10.5194/amt-2022-295, 2022.

Warneke, C., Schwarz, J. P., Dibb, J., Kalashnikova, O., Frost, G., Seidel, F., Al-Saadi, J., Brown, S. S., Washenfelder, R., Brewer, A., Moore, R. H., Anderson, B. E., Yacovitch, T., Herndon, S., Liu, S., Jaffe, D., Johnston, N., Selimovic, V., Yokelson, B., Giles, D., Holben, B., Goloub, P., Popovici, I., Trainer, M., Pierce, B., Fahey, D., Roberts, J., Soja, A., Peterson, D., Saide, P. E., Holmes, C., Wang, S., Coggon, M. M., Decker, Z. C. J., Ye, X., Stockwell, C., Xu, L., Gkatzelis, G., Lefer, B., Crawford, J. Fire Influence on Regional to Global Environments and Air Quality (FIREX-AQ). To be submitted. 2022

Wolfe, G. M., Hanisco, T. F., Arkinson, H. L., Blake, D. R., Wisthaler, A., Mikoviny, T., Ryerson, T. B., Pollack, I., Peischl, J., Wennberg, P. O., Crounse, J. D., St. Clair, J. M., Teng, A., Huey, L. G., Liu, X., Fried, A., Weibring, P., Richter, D., Walega, J., Hall, S. R., Ullmann, K., Jimenez, J. L., Campuzano-Jost, P., Bui, T. P., Diskin, G., Podolske, J. R.,

Sachse, G., and Cohen, R. C.: Photochemical evolution of the 2013 California Rim Fire: synergistic impacts of reactive hydrocarbons and enhanced oxidants, Atmospheric Chem. Phys., 22, 4253–4275, https://doi.org/10.5194/acp-22-4253-2022, 2022.

---

## Author Comment (AC3)

We thank the reviewers for their time and constructive comments that have improved our manuscript. Below we include specific responses to the reviewer's comments. The reviewer's comments are in black. Authors' responses are in blue, quotes from the manuscript are in *italic*, and changes to the text are shown in red.
* * *
Bourgeois et al. presented comprehensive intercomparisons of airborne NO, NO2, HONO, NOy and CO in biomass burning plumes, each measured with differing techniques during FIREX-AQ in the summer of 2019. This study provides valuable dataset and the evaluation of accuracies of major techniques deployed in the challenging biomass burning plume conditions. Additional literature review on these species from major airborne field campaigns are helpful for understanding the accuracy of these measurements under different environmental conditions. The manuscript was written thoroughly, and the figures are made clear. Thus I recommend acceptance after revision. Below are my comments:

1.  Line 204, hourly calibration of NO LIF was performed with [NO] 4-20 ppbv, did this concentration range apply for all the smoke conditions? How do you ensure the linear response beyond this range?

Response: We thank the reviewer for raising this point. As discussed in Rollins et al (2020), given the sensitivity typically observed during FIREX, nonlinearity associated with saturation of the LIF instrument is not problematic until mixing ratios well above 100 ppbv are encountered.
Additionally, linear response to mixing ratios up to 100 ppb has been tested in the laboratory.

We clarified this point in the manuscript lines 217-220:

"*As discussed in Rollins et al (2020), given the sensitivity typically observed during FIREX, nonlinearity associated with saturation of the LIF instrument is not problematic until mixing ratios well above 100 ppbv are encountered.*"

2.  Lines 508-510, "Trajectories and ages that were grossly inconsistent with smoke transport patterns seen in geostationary satellite images were excluded from further analysis". Which group should these data categorized into.

Response: This question is not clear. As stated in the manuscript, these data were simply excluded.

3.  Lines 648-649, what is the p-value of Figure S4 b and d, any explanation for the seemingly dependence of the difference on NO2 concentration?

Response: Few data points actually contributed to the pattern identified by the reviewer. We realized that this figure may be misleading and we replaced it with a new figure (now Figure S6) showing the same plots but color coded by data density.

[Figure]

**Figure S6** Measurement differences (1Hz data) of a) NO, b)–d) NO$_2$, e) HONO, f) NO$_y$, g) CO as a function of the species mixing ratios for the entire campaign. The color bar indicates the number of individual data points per bin of mixing ratios (bin size is 2.5×2.5 ppbv).

4. Figure 3 and Figure 4, no letter label (e.g., a to e) was assigned to any of the panel.

Response: Fixed.

5. In section 3.3.1, intercomparison between CES and CIMS measured HONO were presented. I have the following questions:    1) The slopes shown in Figure 8 suggests CES HONO was higher than CIMS HONO. However, it seems neither the flight averages of the absolute difference shown in Figure S9, nor the histograms of the absolute difference between the two methods suggest the CES-HONO > CIMS-HONO. Any explanation?    2) In Figure S9, why are there many missing points for intercepts (middle panel) and slopes (bottom panel), while the top panel (mean absolute difference) shows all the data on each sampling day?    3) it is interesting to see the measurement of HONO with CIMS are significantly affected by temperature, especially above 30°C, as is shown in Figure S10. Would the slope of CES-HONO vs CIMS-HONO be closer to 1 since it's not shown in this figure?    4) Could the inlets for the two methods be an issue that cause the discrepancy during FIREX-AQ?

Response: We thank the reviewer for raising these points.

1) As stated in the manuscript lines 581-585, the regression analysis (as presented in Figure 8) yields slightly different information than the calculation of the difference: while the former is weighted more by fire plumes, where mixing ratios were greatest, the latter is weighted more by background conditions, where most of the measurements took place. In background conditions, HONO mixing ratios were typically lower than the precision of the CES measurement, yielding a $\Delta HONO_{CES-CIMS}$ close to 0 on average (as reflected in Figures S9 (now Figure S11) and 5e).

2) We thank the reviewer for catching this. We fixed this issue in Figure S9 (now Figure S11) and in the other similar figures as well.

3) In Figure S10 (now Figure S12), the slope is closer to 1 at lower temperatures (slope of 1.3 at 33°C) than at higher temperature (slope of 3.9 at 38°C). A full description of the correction applied to the CIMS HONO data is provided in a follow-up paper that has been submitted to *AMTD* (Robinson et al., 2022)

4) Inlets are unlikely to be the issue. Please see responses above and note that the temperature dependence of the IMR as documented in a separate publication (Robinson et al. 2022) explains the difference without need to invoke inlet effects.

6. Do the measurements shown in Figure 10 (a) include both fresh smoke and aged smoke? If so, what if the fresh smoke and aged smoke were separately considered? Will the relative contribution of each NOy be significantly different? Are the large uncertainties associated with NO2, APNs and pNO3- driven by flight-to-flight difference, secondary processing, or environmental conditions (humidity and temperature)? What could be possible causes for the different contributions of major species (e.g. NO2, APNs and pNO3-) between western wildfires and eastern agriculture fires?

Response: Measurements in Figure 10 include both fresh and aged smoke. The separation of aged and fresh smoke and associated $NO_y$ budget is presented in Figure S11 (now Figure S13). The large range of contribution of $NO_2$, APNs and $pNO_3$ is largely due to the wide range of photochemical conditions sampled during FIREX-AQ – as mentioned before, Figure 10 includes both aged and fresh smoke. The difference in $NO_y$ budget between wildfires and eastern fires may be due to i) a difference in the photochemical aging of the smoke. Most eastern fires produced a thin and dilute smoke plume that was samples close to the fire whereas wildfires usually produced wide and thick plumes that were sampled both close to and further away from the fire; ii) a difference in the fuel. Eastern fires typically consisted of burned crops whereas wildfires fuel consisted of trees and grass.

7. In Figure S11(a), from the slopes determined for fresh versus aged smoke, can we say the sum of NOy outweigh CL-NOy for fresh smoke and the CL-NOy outweigh the sum of NOy, although the difference is within the combined instrumental uncertainties? If so what would the explanation be?

Response: The main difference in the $NO_y$ budget between aged and fresh smoke is that $pNO_3$ becomes the main component of $NO_y$ in aged smoke (Figure S13). Therefore, higher $\Sigma NO_y$ than measured $NO_y$ in aged smoke may be explained by the non-quantitative sampling of $pNO_3$ in the $NO_y$ instrument, as detailed in the section S1 of the SI. In fresh smoke, $pNO_3$ is a smaller component of $NO_y$, and non-quantitative sampling of $pNO_3$ in the CL instrument may

have less impact on the comparison. We added a sentence reflecting this discussion in the text lines 820-827.

*"The variability in the ΣNO_y to NO_y correlation slope between aged and fresh smoke (Figure S13a) likely illustrates the non-quantitative sampling of pNO₃ in the NO_y instrument. Indeed, higher ΣNO_y than measured NO_y in aged smoke (slope of 1.05), where pNO₃ is one of the main components of ΣNO_y (Figure S13b), may be explained by the non-quantitative sampling of pNO₃ in the NO_y instrument. In fresh smoke, pNO₃ is a smaller component of NO_y, and non-quantitative sampling of pNO₃ in the CL instrument may have less impact on the comparison (slope of 0.98)."*

8. Lines 732-734 described what different NOy measurements were used to calculate total NOy. While I understand the choices are based on precision, I wonder why CIMS HONO instead of CES HONO was chosen, as CIMS HONO underestimated CES HONO and its accuracy seems to be significantly affected by temperature variation as is discussed in 3.3.1?

Response: The CIMS HONO measurements were used because they have much better precision than the CES HONO. Precision matters more than accuracy for plots like Figure 5, that are more weighted by background data than by smoke data. Using CES HONO rather than CIMS HONO only affects the slope of the correlation between measured $NO_y$ and $\Sigma NO_y$ by 6%. We clarified this aspect lines 370-372.

*"Using LIF NO, CES NO₂ and CES HONO as primary measurements changed the correlation slope between ΣNO_y and measured NO_y by -2%, -6% and 6%, respectively (Table S1)."*

9. Lines 747-779 are difficult to follow. Figure 12(a) should be well explained first followed by Figure 12 (b). The current order is reversed, and I don't quite get the idea of Figure 12 (a). For Figure 12 (b), it is unclear how the missing NOy fractions (bottom panel) were calculated. My understanding is that fraction of each individual NOy to total NOy was calculated from the individual measurements and sum of NOy, then particle sampling fraction was calculated from the model. Combining the two pieces will enable the quantification of missing NOy (0-24%) resulting from the CL-technique, but how? Thus, further clarification will be needed. Also, in section 3.4.1, it is interesting to see the possible reasons that cause the negative and positive mode of the discrepancy between CL-NOy and sum of NOy. The authors separated the two modes and interpreted them separately. However, if one reason is important (e.g. pNO3- loss through the CL inlet), it should be important throughout the entire campaign, instead of certain period. I might miss something, but a clarification would be helpful.

Response: The reference to Figure 12a (line 774 of the previous version of the manuscript – now line 840) was actually a typo and should have been Figure 12b. Now the discussion first discusses Figure 12a, then Figure 12b.

We calculate the missing NOy according to the following equation:

Missing $NO_y = ((1 - \text{particle sampling fraction}) \times pNO_3) / NO_y$

We added that equation to the text lines 828-835 to clarify the calculation of missing $NO_y$.

"*We calculated the fraction of measured $NO_y$ in smoke initially attributed to $pNO_3$ that may result from other reactive nitrogen species than those included in the $\Sigma NO_y$ according to equation 3:*

*Missing $NO_y$ fraction = $((1 - particle\ sampling\ fraction) \times pNO_3) / NO_y$ (Eq. 3)*

*Where particle sampling fraction corresponds to the modelled $pNO_3$ sampling fraction in the $NO_y$ inlet. We found that missing $NO_y$ accounted for 0–24% of the measured $NO_y$ in smoke (assuming a sampled air speed 65% that of the aircraft; Figure 12b).*"

We respectfully disagree with the reviewer that $pNO_3$ loss through the inlet should be important through the entire campaign. As stated in the manuscript lines 812-813, "*Particle sampling through the $NO_y$ inlet is highly dependent on altitude, air speed (see section S1 and Figure SB) and $pNO_3$ mass size distribution (Figure 12a)*". Also, see our previous response to comment #7 on the effect of aged vs fresh smoke on $NO_y$ closure and clarifications added to the text.

10. In section 3.5.1, it was noted the cause of the discrepancy between ICOS and TDLAS measured CO was unclear. I am curious whether temperature plays a role? Additionally, Figure 14(a) shows when CO goes above 10 ppmv, ICOS seems to outweigh TDLAS; as CO is higher the deviation from 1:1 line is larger. What are the possible explanations?

Response: We interpret this effect as the ICOS having a slower time response than the TDLAS instrument, which is most noticeable when the mixing ratio is high and the plume width is narrow.

**References:**

Robinson, M. A., Neuman, J. A., Huey, L. G., Roberts, J. M., Brown, S. S., and Veres, P. R.: Temperature dependent sensitivity of iodide chemical ionization mass spectrometers, EGUsphere, 1–17, https://doi.org/10.5194/amt-2022-295, 2022.